# Flexible Neural Image Compression via Code Editing

**Chenjian Gao**[*1], **Tongda Xu**[*1,2], **Dailan He**[1], **Hongwei Qin**[1], **Yan Wang**[1,2] [†]
[1]SenseTime Research, [2]Institute for AI Industry Research (AIR), Tsinghua University
{xutongda, wangyan}@air.tsinghua.edu.cn,
{gaochenjian, hedailan, qinhongwei}@sensetime.com

## Abstract

Neural image compression (NIC) has outperformed traditional image codecs in rate-distortion (R-D) performance. However, it usually requires a dedicated encoder-decoder pair for each point on R-D curve, which greatly hinders its practical deployment. While some recent works have enabled bitrate control via conditional coding, they impose strong prior during training and provide limited flexibility. In this paper we propose Code Editing, a highly flexible coding method for NIC based on semi-amortized inference and adaptive quantization. Our work is a new paradigm for variable bitrate NIC, and experimental results show that our method surpasses existing variable-rate methods. Furthermore, our approach is so flexible that it can also achieves ROI coding and multi-distortion trade-off with a single decoder. Our approach is compatible to all NIC methods with differentiable decoder NIC, and it can be even directly adopted on existing pre-trained models.

## 1 Introduction

Lossy image compression is a fundamental problem of computer vision. In a simplified setting, the aim of lossy compression is to minimize rate-distortion cost $R + \lambda D$, with $\lambda$ as the Lagrange multiplier controlling R-D trade-off. Traditional compression algorithms (e.g. JPEG, JPEG2000 and BPG [Bellard, 2018]) first perform a linear transformation (e.g. DCT, wavelet transform) and then quantize the transformed coefficients for compression. Their bitrate control is achieved by controlling quantization stepsize. On the other hand, lossy neural image compression (NIC) methods have outperformed traditional codecs in recent years [Xie et al., 2021, He et al., 2022]. These methods implement powerful non-linear transformation with a neural network, and therefore achieve better R-D performance. However, in order to realize R-D trade-offs, we need to optimize multiple networks towards different losses with different $\lambda$s, which means that we cannot achieve flexible bitrate control.

To enable NIC models with bitrate control ability, several works adopt conditional networks [Choi et al., 2019, Cui et al., 2021, Sun et al., 2021] . They use $\lambda$ as condition and feed it into the encoder and decoder. During training, they sample $\lambda$ from a pre-defined prior and optimize the $\mathbb{E}_{p(\lambda)}[R + \lambda D]$ via Monte Carlo. Due to the capacity of conditional networks, most of them use simple prior and limited sets of $\lambda$, such as a uniform distribution over only $5 - 10$ discrete $\lambda$s [Choi et al., 2019, Cui et al., 2021]. And additional effort is required to interpolate the discrete $\lambda$s to achieve continuous R-D curve [Choi et al., 2019, Cui et al., 2021, Sun et al., 2021], which limits the flexibility of variable bitrate model. On the other hand, traditional image codecs achieve flexible R-D trade-off naturally by controlling the quantization step. This control is fine grain (e.g. 100 levels for JPEG). Moreover, they support flexible spatial bits-allocation for ROI-based (region of interest) coding. And the user need only one decoder for all those flexibility.

---

[*]Equal contributions.
[†]Corresponding author.

36th Conference on Neural Information Processing Systems (NeurIPS 2022).

In this work, we intend to give neural image compression methods the flexibility of traditional codecs: the continuous rate-distortion trade-off with arbitrary spatial bits-allocation by one decoder. It is a meaningful task for practical NIC considering the growing size of neural decoders and the need for fine-grain rate control. In order to achieve this, we propose Code Editing, a semi-amortized inference [Kim et al., 2018] based approach for flexible R-D trade-off. Although previous works have adopted semi-amortized inference to improve R-D performance [Johnston et al., 2018, Yang et al., 2020], we are the first to discover its potential for flexible bitrate control. Specifically, we edit the latent code directly to change the optimization target at inference time based on desired $\lambda$s and ROIs. However, we find that naïve Code Editing has limited bitrate control ability. We further analyze and address the problem via quantization step size adaptation, and enhance its speed-performance trade-off. Surprisingly, we find that our Code Editing can achieve not only flexible continuous bitrate control and ROI-based coding, but also multi-distortion trade-off, all with one decoder.

To wrap up, our contributions are as follows:

- We propose Code Editing, a new paradigm for variable bitrate NIC based on semi-amortized inference. It supports continuous bitrate control with a single decoder and outperforms previous variable bitrate NIC methods. To the best of knowledge, we are the first to consider semi-amortized inference for variable bitrate NIC.
- We resolve the performance decay of Code Editing by combining it with adaptive quantization step size. And we further improve the speed-performance trade-off of Code Editing.
- We demonstrate the potential of Code Editing in flexible spatial bits-allocation and multi-distortion trade-off, which is beyond the flexibility of traditional codecs.

## 2 Code Editing: Flexible Neural Image Compression

### 2.1 Background

The general optimization procedure of NIC methods can be concluded as follows: 1) given an image $\boldsymbol{x}$, an encoder (inference model) produces latent parameter $\boldsymbol{y} = f_\phi(\boldsymbol{x})$. 2) Then, we obtain $\lfloor \boldsymbol{y} \rceil$ by unit scalar quantization. 3) Then, an entropy model (prior) $p_\theta(\boldsymbol{y})$ parameterized by $\theta$ is used to compute the probability mass function (pmf) $P_\theta(\lfloor \boldsymbol{y} \rceil) = \prod \int_{\lfloor \boldsymbol{y}^{(i)} \rceil - 0.5}^{\lfloor \boldsymbol{y}^{(i)} \rceil + 0.5} p_\theta(\boldsymbol{y}^{(i)}) d\boldsymbol{y}^{(i)} = \prod(F_\theta(\lfloor \boldsymbol{y}^{(i)} \rceil + 0.5) - F_\theta(\lfloor \boldsymbol{y}^{(i)} \rceil + 0.5))$, where $F_\theta$ is the cdf of $p_\theta$. 4) With that pmf, we encode $\lfloor \boldsymbol{y} \rceil$ with bitrate $R = -\log P_\theta(\lfloor \boldsymbol{y} \rceil)$ by entropy coder. 5) For the decoding process, $\lfloor \boldsymbol{y} \rceil$ is feed into decoder network to obtain reconstruction image $\bar{\boldsymbol{x}} = g_\theta(\lfloor \boldsymbol{y} \rceil)$, and $d(\cdot, \cdot)$ is used to compute the distortion between reconstruction and original image. The distortion $D$ can be interpreted as the likelihood $\log p(\boldsymbol{x}|\lfloor \boldsymbol{y} \rceil)$ so long as we treat distortion as energy of Gibbs distribution [Minnen et al., 2018]. For example, when $d(\cdot, \cdot)$ is pixel-wise MSE, we can interpret $d(\boldsymbol{x}, \bar{\boldsymbol{x}}) = -\log p(\boldsymbol{x}|\lfloor \boldsymbol{y} \rceil) + constant$, where $p(\boldsymbol{x}|\lfloor \boldsymbol{y} \rceil) = \mathcal{N}(\bar{\boldsymbol{x}}, 1/2\lambda I)$. 6) Finally, the encoder parameter $\phi$, decoder and entropy model parameter $\theta$ are optimized to minimize the R-D cost $R + \lambda D$, where $R = -\log P_\theta(\lfloor \boldsymbol{y} \rceil)$ and $D = d(\boldsymbol{x}, g_\theta(\lfloor \boldsymbol{y} \rceil))$. This procedure can be described by Eq. 1 and Eq. 2. As the rounding operation $\lfloor \cdot \rceil$ is non-differentiable, the majority works of NIC adopt additive uniform noise (AUN) to relax it [Ballé et al., 2017, 2018, Cheng et al., 2020].

$$\theta^*, \phi^* = \arg\max_{\theta, \phi} \mathcal{L}_{\theta,\phi} \tag{1}$$

$$\mathcal{L}_{\theta,\phi} = -(R + \lambda D) = \mathbb{E}_{p(\boldsymbol{x})}[\underbrace{\log P_\theta(\lfloor \boldsymbol{y} \rceil)}_{\text{-rate}} - \lambda \underbrace{d(\boldsymbol{x}, g_\theta(\lfloor \boldsymbol{y} \rceil))}_{\text{distortion}}] \tag{2}$$

With the AUN relaxed latent code $\tilde{\boldsymbol{y}} = \boldsymbol{y} + \mathcal{U}(-0.5, 0.5)$, the encoding and decoding of NIC can be formulated as a Variational Autoencoder (VAE) [Kingma and Welling, 2013] on a probabilistic graphic model $\tilde{\boldsymbol{y}} \rightarrow \boldsymbol{x}$, where $\tilde{\boldsymbol{y}}$ are continuous relaxed latent codes. The prior likelihood $\log p_\theta(\tilde{\boldsymbol{y}})$ of such VAE is a continuous relaxation of $\log P_\theta(\lfloor \boldsymbol{y} \rceil)$. The data likelihood $\log p_\theta(\boldsymbol{x}|\tilde{\boldsymbol{y}})$ is the distortion offseted by a constant. Moreover, the AUN relaxed latent code $\tilde{\boldsymbol{y}}$ can be interpreted as the reparameterized sample through a factorized uniform posterior $q_\phi(\tilde{\boldsymbol{y}}|\boldsymbol{x}) = \mathcal{U}(\boldsymbol{y} - 0.5, \boldsymbol{y} + 0.5)$. Under such formulation, the evidence lower bound (ELBO) is directly connected to the negative R-D cost of compression (Eq. 3). Then, minimizing $R + \lambda D$ is relaxed into maximizing $\mathcal{L}_{\theta,\phi}^{train}$.

$$\mathcal{L}_{\theta,\phi}^{train} = \mathbb{E}_{p(\boldsymbol{x})}[\mathbb{E}_{q_\phi(\tilde{\boldsymbol{y}}|\boldsymbol{x})}[\underbrace{\log p_\theta(\tilde{\boldsymbol{y}})}_{\text{- rate}} + \underbrace{\log p_\theta(\boldsymbol{x}|\tilde{\boldsymbol{y}})}_{\text{- distortion}} - \underbrace{\log q_\phi(\tilde{\boldsymbol{y}}|\boldsymbol{x})}_{0}]] \tag{3}$$

The above process of encoding is also known as fully amortized variational inference, as for each new image $\boldsymbol{x}'$, its variational posterior is fully determined by amortized paramters $\phi$. Those parameters are called amortized parameters as they are global parameters shared by all data points.

## 2.2 Code Editing Naïve

Different from previous methods in variable bitrate NIC, we propose a new paradigm of controlling R-D trade-off by semi-amortized inference, named Code Editing. Consider now we have a pair of parameters $\theta_{\lambda_0}, \phi_{\lambda_0}$ optimized for a specific R-D trade-off parameter $\lambda_0$. Then, for a new Lagrangian multiplier $\lambda_1$, we encode by editing the code. To be specific, given an image $\boldsymbol{x}$, we first initialize the unquantized latent parameters $\boldsymbol{y} \leftarrow f_{\phi_{\lambda_0}}(\boldsymbol{x})$. Note that now $\boldsymbol{y}$ is exactly the same as the latent with R-D trade-off $\lambda_0$. Next, we iteratively optimize $\boldsymbol{y}$ to maximize the target $\mathcal{L}_{\boldsymbol{y}}$ as Eq. 5. In other words, we directly edit the code $\boldsymbol{y}$ to adapt to different R-D trade-offs. During the optimization process, the decoder and entropy parameter $\theta_{\lambda_0}$ remains the same. This process of encoding is known as semi-amortized inference [Kim et al., 2018], as it utilizes the initial value of amortized inference, and optimizes latent for each data point afterwards.

$$\boldsymbol{y}^* = \arg\max_{\boldsymbol{y}} \mathcal{L}_{\boldsymbol{y}} \tag{4}$$

$$\mathcal{L}_{\boldsymbol{y}} = -(R + \lambda_1 D) = \log P_{\theta_{\lambda_0}}(\lfloor \boldsymbol{y} \rceil) - \lambda_1 d(\boldsymbol{x}, g_{\theta_{\lambda_0}}(\lfloor \boldsymbol{y} \rceil)) \tag{5}$$

Similar to NIC, one challenge remains is that the rounding operation $\lfloor . \rceil$ is not-differentiable. It is not possible to adopt gradient based optimization approach directly. Learning from Yang et al. [2020], we adopt stochastic gumbel annealing (SGA) in lieu of rounding to obtain a surrogate ELBO. As shown in Sec. 4.2, adopting SGA here achieves better R-D performance than AUN. After that, we use standard gradient based optimization methods with temperature annealing to optimize latent $\boldsymbol{y}$. And we call this method Code Editing Naïve.

## 2.3 Code Editing Enhanced

Empirically, we find that Code Editing Naive can only achieve continuous R-D trade-off within a narrow range (e.g. $\pm 0.1$ bpp). Out of this range, the R-D performance falls rapidly (See Sec. 4.2). One possible reason is the train-test mismatch of the entropy model $p_{\theta_{\lambda_0}}(\boldsymbol{y})$. Naturally, $\lfloor \boldsymbol{y}^* \rceil$ grows sparser as bitrate decreases. And this causes an obvious gap between latent distribution optimized for $\lambda_0$ and $\lambda_1$ (See Fig. 2). Thus, the entropy model fitted to latent distribution at $\lambda_0$ might perform poorly in estimating latent density of other $\lambda$s. More specifically, the mismatched bitrate is $\mathbb{E}_{p(\boldsymbol{x})}[\mathbb{E}_{q(\boldsymbol{y}^*|\boldsymbol{x})}[\log P_{\theta_{\lambda_1}}(\lfloor \boldsymbol{y}^* \rceil) - \log P_{\theta_{\lambda_0}}(\lfloor \boldsymbol{y}^* \rceil)]]$. Under the assumption that the variational posterior is the true posterior, this mismatch equal to $D_{KL}(P_{\theta_{\lambda_1}}(\lfloor \boldsymbol{y}^* \rceil) || P_{\theta_{\lambda_0}}(\lfloor \boldsymbol{y}^* \rceil))$, which is just the distance between two distributions.

In NIC, the probability mass function (pmf) $P_\theta(\lfloor \boldsymbol{y} \rceil)$ over quantized $\boldsymbol{y}$ is computed by taking the difference of cumulative distribution function. In other words, $P_{\theta_{\lambda_0}}(\lfloor \boldsymbol{y} \rceil) = \prod(F_{\theta_{\lambda_0}}(\lfloor \boldsymbol{y}^{(i)} \rceil + 0.5) - F_{\theta_{\lambda_0}}(\lfloor \boldsymbol{y}^{(i)} \rceil - 0.5))$. Following Choi et al. [2019], we augmenting it with the quantization step $\Delta$, and this results in $P_{\theta_{\lambda_0}}(\lfloor \boldsymbol{y}/\Delta \rceil; \Delta) = \prod(F_{\theta_{\lambda_0}}(\Delta \lfloor \boldsymbol{y}^{(i)}/\Delta \rceil + \Delta/2) - F_{\theta_{\lambda_0}}(\Delta \lfloor \boldsymbol{y}^{(i)}/\Delta \rceil - \Delta/2))$, where $\Delta \lfloor \cdot /\Delta \rceil$ means the dequantized result of quantization with step $\Delta$. Controlling $\Delta$ results in significantly different pmf without changing the underlying continuous density $p_{\theta_{\lambda_0}}(\boldsymbol{y})$. Then, we can optimize the $\Delta$ augmented ELBO to find the optimal code $y^*$ and quantization step size $\Delta^*$ as Eq. 6 and Eq. 7.

$$\boldsymbol{y}^*, \Delta^* = \arg\max_{\boldsymbol{y},\Delta} \mathcal{L}_{\boldsymbol{y},\Delta} \tag{6}$$

$$\mathcal{L}_{\boldsymbol{y},\Delta} = -(R + \lambda_1 D) = \log P_{\theta_{\lambda_0}}(\lfloor \boldsymbol{y}/\Delta \rceil; \Delta) - \lambda_1 d(\boldsymbol{x}, g_{\theta_{\lambda_0}}(\Delta \lfloor \boldsymbol{y}/\Delta \rceil)) \tag{7}$$

In fact, similar idea of controlling $\Delta$ has been adopted in Choi et al. [2019]. It implements continuous R-D trade-off within small bitrate range (e.g. $\pm 0.2$ bpp) by controlling $\Delta$. However, we show that a large range of continuous R-D trade-off (e.g. $0.1 - 1.1$ bpp) can be achieved by combining the quantization step size control and semi-amortized inference.

## 2.4 Speed-Performance Trade-off

Optimizing $y$ over Eq. 7. from $f_{\phi_0}(x)$ for each $x$ is time-consuming. Naturally, we can accelerate Code Editing by reducing the iteration of optimization, and the R-D performance would drop accordingly. In practice, we find that reducing the number of iterations to $10\%$ of full convergence achieves reasonable results for low bitrate regions (with 0.4 db PSNR drop). However, the R-D performance in the high bitrate range degrades severely. We propose to enhance the amortized encoder by finetuning at a high bitrate R-D objective. Then the finetuned encoder can predict a better initial $y$ for a high bitrate range, and thus makes computationally scalable Code Editing possible.

## 2.5 Extension of Code Editing

With code editing, we can empower NIC with more flexible coding capabilities out of continuous bitrate control. By extending the optimization target for code editing, we can achieve different coding requirements. In this work, we explore the potential of code editing in ROI-based coding and multi-distortion trade-off.

**ROI-based Coding.** The need of ROI-based coding stems from the fact that different pixels in an image have different levels of importance. So that spatially different bitrate should be assigned according to regions of interest (ROI) during compression. When performing ROI-based coding, a quality map $m \in \mathbb{R}^{H,W}$ is provided to the encoder in addition to the input image $x$, which indicates the importance of each pixel. Here we use a bounded continuous-valued quality map within $[0, 1]$, where 0 and 1 represent the least and the most important pixel respectively. To make Code Editing support ROI-based coding, we extent the objective in Eq. 7. Specifically, we weight the per-pixel distortion with ROI map $m$ to obtain the optimization target $\mathcal{L}^{ROI}_{y,\Delta}$, where $\circ$ represents element-wise product.

$$\mathcal{L}^{ROI}_{y,\Delta} = \log P_{\theta_{\lambda_0}}(\lfloor y/\Delta \rceil; \Delta) - \lambda_1 \underbrace{m \circ d(x, g_{\theta_{\lambda_0}}(\Delta\lfloor y/\Delta \rceil))}_{\text{ROI weighted distortion}} \tag{8}$$

Note that different from existing ROI-based coding works for NIC [Song et al., 2021], our approach imposes no prior on the shape of ROI during training. And thus our ROI control is more flexible (See results in Sec. 4.5)

**Multi-Distortion Trade-off.** When compressing an image, sometimes we want to optimize the MSE, sometimes we want to optimize the perceptual loss, and sometimes we want a balanced trade-off between them. However, the different distortion metrics are in odds to each other [Blau and Michaeli, 2018, 2019]. For conventional NIC, we can add multiple distortion term in Eq. 7 to optimize $\theta, \phi$ for a specific weights between rate, and different distortions. However, just like R-D trade-off, multi-distortion trade-off also requires multiple, or even infinite number of decoders to achieve.

Again, our Code Editing can achieve the multi-distortion trade-off by extending the objective in Eq. 7. For example, we we want to balance the distortion and perceptual quality, we optimize towards $R + \lambda_d D_d + \lambda_p D_p$, where $D_d = d_d(x, g_{\theta_{\lambda_0}}(\Delta\lfloor y/\Delta \rceil))$ is MSE, $D_p = d_p(x, g_{\theta_{\lambda_0}}(\Delta\lfloor y/\Delta \rceil))$ is the LPIPS [Zhang et al., 2018] and $\lambda_d, \lambda_p$ are parameters controlling the trade-off. The extended target is as Eq. 9. Similarly, we can achieve this target by optimizing $y$ and $\Delta$ to maximize $\mathcal{L}^{MD}_{y,\Delta}$.

$$\mathcal{L}^{MD}_{y,\Delta} = \log P_{\theta_{\lambda_0}}(\lfloor y/\Delta \rceil; \Delta) - \lambda_d \underbrace{d_d(x, g_{\theta_{\lambda_0}}(\Delta\lfloor y/\Delta \rceil))}_{\text{distortion loss}} - \lambda_p \underbrace{d_p(x, g_{\theta_{\lambda_0}}(\Delta\lfloor y/\Delta \rceil))}_{\text{perceptual loss}} \tag{9}$$

Moreover, it is theoretically possible to achieve a trade-off between multiple loss metrics by extending the target as Eq. 10. For example, we can let $d_1$ be the MSE , $d_2$ be the VGG loss, $d_3$ be the style loss. Then, we can optimize our latent code to achieve the sophisticated photo-realistic effect described in Sajjadi et al. [2017]. And it is so flexible that we do not even need to know what those loss functions

are during training. However, we have not conducted empirical study on $\mathcal{L}_{\boldsymbol{y},\Delta}^{MUL}$ as it is a much less common scenario in practice.

$$\mathcal{L}_{\boldsymbol{y},\Delta}^{MUL} = \log P_{\theta_{\lambda_0}}(\lfloor \boldsymbol{y}/\Delta \rceil; \Delta) - \sum \lambda_i d_i(\boldsymbol{x}, g_{\theta_{\lambda_0}}(\Delta \lfloor \boldsymbol{y}/\Delta \rceil)) \tag{10}$$

### 2.6 Hierarchical Latent Case

Most of the sota NIC methods [Minnen et al., 2018, Minnen and Singh, 2020, Cheng et al., 2020, He et al., 2022] are based on the hierarchical latent framework proposed by Ballé et al. [2018], which has graphic model $\tilde{\boldsymbol{z}} \to \tilde{\boldsymbol{y}} \to \boldsymbol{x}$. In this framework, $\tilde{\boldsymbol{y}}, \tilde{\boldsymbol{z}}$ are relaxed latent code and $\boldsymbol{x}$ is image. To simplify notation, in this section we base our analysis upon 1 level latent framework by Ballé et al. [2017] with graphic model $\tilde{\boldsymbol{y}} \to \boldsymbol{x}$. We note that all the formulas in this section can be easily extended to the hierarchical latent framework. We also note that for hierarchical latent case, the $\boldsymbol{y}$'s quantization stepsize $\Delta_y$ is directly optimized by gradient descent, and the $\boldsymbol{z}$'s quantization stepsize $\Delta_z$ is optimized by grid search (See Appendix A.2 and A.3).

## 3 Related Works

### 3.1 Variable Bitrate and ROI-based control for NIC

The pioneers of NIC [Toderici et al., 2016, 2017, Johnston et al., 2018] achieve coarse rate control by growing the number of residuals. On the other hand, Theis et al. [2017] learns the scaling vector of latent code for each bitrate to control the bitrate. However, their R-D performance is surpassed by Ballé et al. [2017]. More recently, based on the framework by Ballé et al. [2017], Choi et al. [2019], Cui et al. [2021], Sun et al. [2021] uses the one-hot encoded $\lambda$ as conditional input to the auto-encoder where $\lambda$ is sampled from uniform discrete prior during training. And additional efforts are required to interpolate the R-D curve for continuous rate control. Cui et al. [2021] improves Theis et al. [2017] with asymmetrically scaled vectors. More recently, Song et al. [2021] inserts Spatial Feature Transform (SFT) module into the auto-encoder to modulate the intermediate feature. However, to the best of knowledge, we are the first to explore semi-amortized inference based variable bitrate model.

For ROI-based control, Agustsson et al. [2019] distinguishes images into important and unimportant regions, and use a GAN to generate the unimportant regions. Cai et al. [2020] supports bit allocation by applying different losses to ROI and non-ROI regions. Song et al. [2021] uses ROI to modulate the intermediate features and recover the ROI implicitly from the hyperprior, and it claims that it support flexible ROI control. However, it requires complicated handcrafted ROI prior during training. Compared with above-mentioned methods, our Code Editing does not require any prior of ROI, nor data contains segmentation map during training.

After the submission of this manuscript, Shi et al. [2022], Fathima et al. [2023] also achieve pixel level variable rate coding for neural codec. On the other hand, Agustsson et al. [2022] also implement promising perception-distortion trade-off with a conditional decoder.

### 3.2 Semi-amortized Inference for NIC

The semi-amortized variational inference is concurrently invented by Kim et al. [2018] and Marino et al. [2018]. The basic idea is that works following Kingma and Welling [2013] lose the good tradition of learning variational posterior parameters per datapoint. Instead they learn fully amortized parameters for the whole dataset. And this fully amortized inference might lead to sub-optimal posterior distribution. Cremer et al. [2018] refers to this phenomena as amortization gap and argues it is a major contributor to inference sub-optimality of VAE. The semi-amortized inference is to initialize the variational posterior's parameter with fully amortized parameters, and conduct stochastic optimization per data-point.

Although Kim et al. [2018], Marino et al. [2018] are alluring to apply, they require nested iterative optimization within the stochastic gradient descent loop. When adopting semi-amortized inference to NIC, Djelouah and Schroers [2019], Yang et al. [2020] simplify it into two stages: 1) training a fully amortized encoder-decoder pair 2) iteratively optimize latent which is initialized by amortized encoder. Although this simplification can lead to sub-optimal generative model (recall that now

the generative model is paired with sub-optimal inference model), it indeed makes semi-amortized inference practical for NIC. Our method follows Yang et al. [2020] as we explore semi-amortized inference for R-D trade-off instead of R-D performance.

## 4 Experimental Results

### 4.1 Experimental Settings

Following He et al. [2021], we train the baseline models on a subset of 8,000 images from ImageNet. For training, we use MSE as distortion. For each baseline method, we train models with 7 fixed R-D trade-off points, with $\lambda \in \{0.0016, 0.0032, 0.0075, 0.015, 0.03, 0.045, 0.08\}$. This setup is aligned with Cheng et al. [2020]. The baseline models' bitrate ranges from 0.1 to 1.0 bpp on Kodak dataset [Kodak, 1993]. All baseline models are trained using the Adam optimizer for 2000 epochs. Batchsize is set to 16 and the initial learning rate is set to $10^{-4}$. For all the results reported in the R-D curve, the bitrate is measured by actual bits of range encoder, and the reconstruction is computed by the latent coded from the actual the range decoder. For all visualization involves spatial bitrate distribution, the theoretical bitrate is used. More detailed setup can be found in Appendix C.

### 4.2 Ablation Study

We set Ballé et al. [2018] as the baseline of the ablation study. All experiments involving Code Editing are based on the model trained with $\lambda_0 = 0.015$. In ablation study, all the results are tested on Kodak dataset.

**Adaptive $\Delta$.** To reduce the prior mismatch, we propose to control the quantization step $\Delta$ in addition to optimizing $y$. To verify its effectiveness, we compare Code Editing Enhnanced (w/ adaptive $\Delta$) with Code Editing Naïve (w/o adaptive $\Delta$). As shown in Fig. 1, the R-D performance of Code Editing Naïve drops evidently when the bitrate changes out of a small range ($\pm$ 0.1 bpp). Further, we provide statistics of normalized dequantized results of quantized $y^{(i)}$ before and after Code Editing on the Kodak dataset in Fig. 2. It is shown that the distribution mismatch of Code Editing Naïve is more evident than Code Editing Enhanced. On the other hand, we also show the result of controlling $\Delta$ alone. As Choi et al. [2019], we find that adjusting $\Delta$ can shift the bitrate in a limited range ($\pm$ 0.1 bpp). But out of this range, the R-D performance drops rapidly.

**SGA vs. AUN.** In Code Editing, we adopt SGA instead of additive uniform noise (AUN) as relaxation of discrete latent. To verify the effect of SGA, we optimize $y$ for 2,000 iterations using SGA and AUN, respectively. As shown in Fig. 1, although Code Editing with AUN also outperforms the baseline, the SGA's R-D performance is significantly higher than AUN especially in high bitrate region. This is due to the fact that SGA closes the discretization gap [Yang et al., 2020] via soft to hard annealing.

**Encoder Fine-tuning for Fast Inference.** We show that we can control the computational complexity of Code Editing by early termination. We terminate Code Editing with $\{50, 100, 200\}$ iterations, which correspond to $\{2.5\%, 5\%, 10\%\}$ of full iterations. The results on the Kodak dataset are shown in Fig. 1. In low bitrate range, the R-D performance achieves a reasonable point within $10\%$ of iterations. However, the R-D performance in the high bitrate region degrades severely. To solve this problem, we finetune encoder with $\lambda = 0.052$ under $\Delta_y = 0.5$. The latent code predicted by the finetuned encoder is used as the initial value for Code Editing in the high bitrate range. And a stronger encoder achieves better R-D performance with very few iterations (marked with "Encoder FT." in the Fig. 1).

### 4.3 Variable Rate Coding

First, we evaluate Code Editing Enhanced for continuous variable bitrate coding. We choose three established works [Ballé et al., 2018, Minnen et al., 2018, Cheng et al., 2020] as baselines. For each of the baselines, we set base model's R-D trade-off parameter $\lambda_0 = 0.015$, which results in a mid-range bitrate (around 0.5 bpp). For each image $x$ to be compressed, we initialize $y \leftarrow f_{\phi_{\lambda_0}}(x)$ and optimize $y$ for 2,000 iterations using SGA, with the learning rate of $5 \times 10^{-3}$. This setting is aligned with Yang et al. [2020]. $\Delta_y$ is directly optimized by gradient descent and $\Delta_z$ is optimized by grid search. Fig. 3 shows the R-D performance of variable bitrate via Code Editing. It can be seen

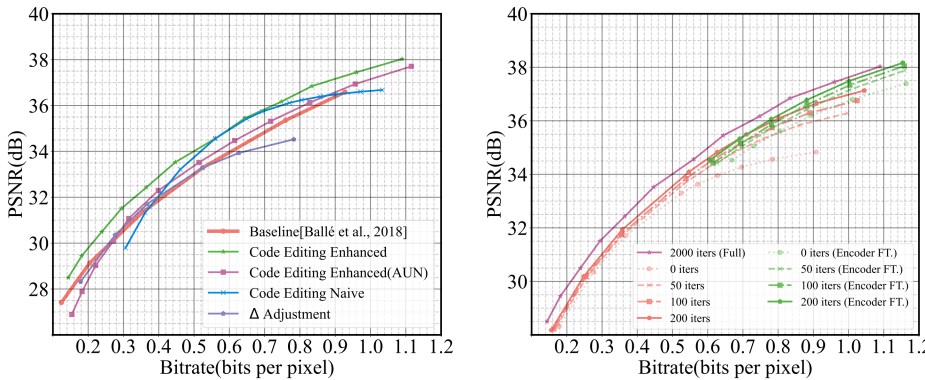

Figure 1: Results of ablation study. Left: Results of adaptive $\Delta$ and SGA. Right: Results of speed-performance trade-off. $\Delta$ adjustment is used for "0 iters".

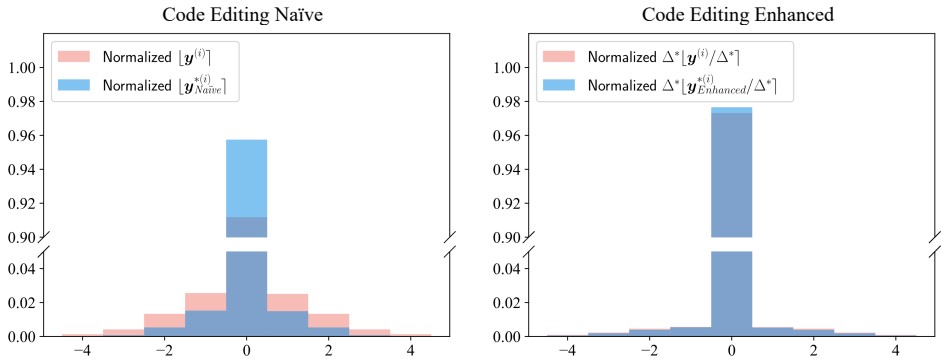

Figure 2: The distribution of normalized dequantized results of quantized $\boldsymbol{y}^{(i)}$ before and after Code Editing. The normalization parameter is $\sigma^2$ from $p(\boldsymbol{y}|\boldsymbol{z})$. Left: Code Editing Naïve. Right: Code Editing Enhanced. The source $\lambda_0 = 0.015$, the target $\lambda_1 = 0.0016$.

that Code Editing Enhanced can effectively achieve wide range continuous R-D trade-off with fixed decoder and entropy model trained at $\lambda_0$.

Then, we compared Code Editing Enhanced with the sota variable code-rate methods [Theis et al., 2017, Cui et al., 2021, Song et al., 2021]. For fairness, we re-implement these methods on the baselines with the same setting as original papers. The results are shown in Fig. 3. Note that we only show the results of Song et al. [2021] on Ballé et al. [2018] as it produces NaNs on other models. As shown in Fig. 3, Our proposed Code Editing outperforms existing variable bitrate methods with

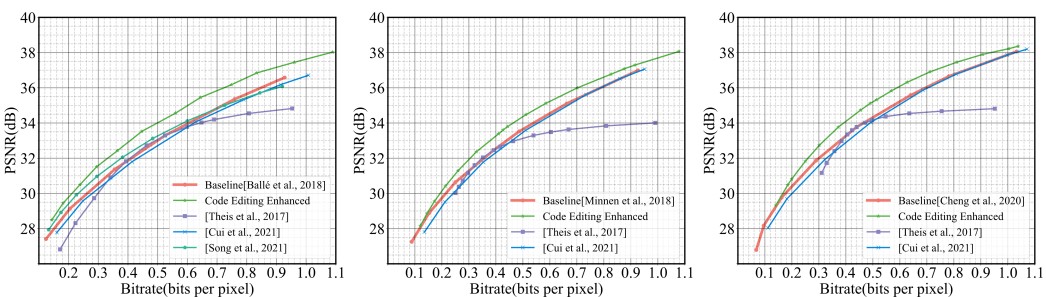

Figure 3: Results of Code Editing Enhanced on Kodak dataset. Left: Ballé et al. [2018] as baseline. Middle: Minnen et al. [2018] as baseline. Right: Cheng et al. [2020] as baseline.

all three baseline. Theis et al. [2017] brings larger degradation in R-D performance, especially in the high bitrate region. Cui et al. [2021] achieves similar R-D performance as the original fix-bitrate model. The R-D performance of Song et al. [2021] on Ballé et al. [2018] exceeds the fix-bitrate model in the low bitrate range but is inferior to Code Editing Enhanced.

## 4.4 Multi-Distortion Trade-off

Next, we explore the potential of Code Editing Enhanced in multi-distortion trade-off. For experiments, we choose to balance between MSE and LPIPS [Zhang et al., 2018]. MSE is adopted to measure fidelity, while LPIPS is widely adopted for perceptual quality [Mentzer et al., 2020, Bhat et al., 2021].The lower the LPIPS is, the better perceptual quality an image has. In our experiment, we select 3 trade-off points by controlling the LPIPS weight $\lambda_p \in \{0.1, 0.5, 1.0\}$. We use the bitrate control scheme in Mentzer et al. [2020] to stabilize bitrate (See Appendix C for detail). Fig. 4 shows the multi-distortion trade-off results based on Ballé et al. [2018] on the Kodak dataset. As $\lambda_p$ increases, both LPIPS and PSNR decrease, implying better perceptual quality and worse distortion is achieved. Further, we show qualitative results based on Ballé et al. [2018] on the CLIC2022 dataset [CLIC, 2022] in Fig. 5. We can see that as $\lambda_p$ increases, the detail level of image increases, which indicates a better perceptual quality. Both quantitative and qualitative results show that our Code Editing can achieve flexible multi-distortion trade-off.

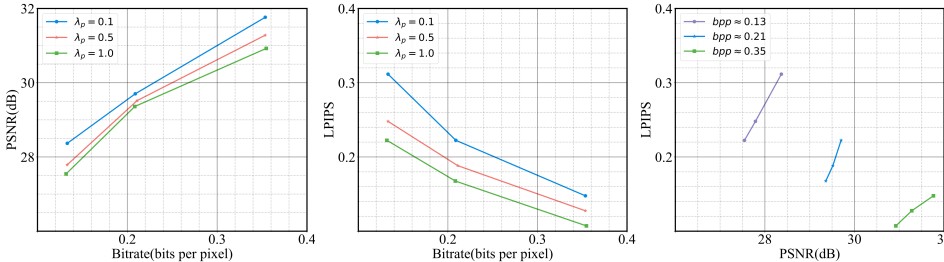

Figure 4: The quantitative multi-distortion trade-off results. Left: Bitrate-PSNR with different $\lambda_p$s. Middle: Bitrate-LPIPS with different $\lambda_p$s. Right: PSNR-LPIPS with different $\lambda_p$s, the bitrate are approximately the same.

## 4.5 ROI-based Coding

This section shows the performance of Code Editing in terms of ROI-based coding. In order to verify the effectiveness of spatially adaptive R-D trade-off, we use various quality maps. We use Ballé et al. [2018] as the baseline model to verify the effectiveness of Code Editing in ROI-based coding on Kodak and CLIC2022 datasets. Fig. 6 illustrates these results. And our Code Editing can allocate bitrate spatially according to arbitrary-shape ROI maps, including checkerboard and alphabet shapes. Moreover, we compared our Code Editing ROI results with Song et al. [2021] which is trained with sophisticated prior in Fig. 7. It can be seen that Song et al. [2021]'s result suffers from ROI map diminishing issue. We can barely sense the effect of ROI map inside the region indicated by the white rectangle. In Appendix B.2, we provide more ROI results on different types of masks, including segmentation mask.

## 4.6 Gap to SAVI with Multiple Models

We present the results of direct performing semi-amortized inference on baseline models on Kodak dataset. To be specific, we directly optimize $y$ using SGA based on multiple models, which is equivalent to Yang et al. [2020] without bits-back coding. From Fig. 8 we can see that when baselines are Ballé et al. [2018], Minnen et al. [2018], our Code Editing Enhanced is comparable to SGA based on multiple models. However, when baseline is Cheng et al. [2020], our Code Editing Enhanced indeed suffers from R-D performance decay in high bitrate region. We use a single decoder to achieve this while Yang et al. [2020] require multiple decoder. We are neither superior nor inferior to Yang et al. [2020], as the task is very different.

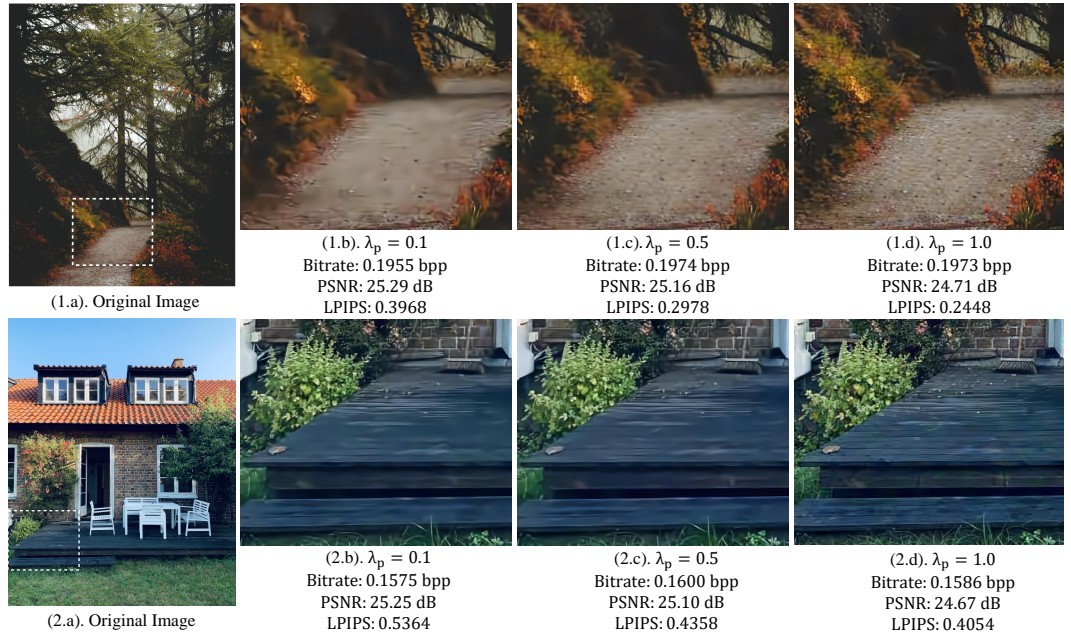

(1.b). $\lambda_p = 0.1$
Bitrate: 0.1955 bpp
PSNR: 25.29 dB
LPIPS: 0.3968

(1.c). $\lambda_p = 0.5$
Bitrate: 0.1974 bpp
PSNR: 25.16 dB
LPIPS: 0.2978

(1.d). $\lambda_p = 1.0$
Bitrate: 0.1973 bpp
PSNR: 24.71 dB
LPIPS: 0.2448

(1.a). Original Image

(2.b). $\lambda_p = 0.1$
Bitrate: 0.1575 bpp
PSNR: 25.25 dB
LPIPS: 0.5364

(2.c). $\lambda_p = 0.5$
Bitrate: 0.1600 bpp
PSNR: 25.10 dB
LPIPS: 0.4358

(2.d). $\lambda_p = 1.0$
Bitrate: 0.1586 bpp
PSNR: 24.67 dB
LPIPS: 0.4054

(2.a). Original Image

Figure 5: The qualitative multi-distortion trade-off results. (x.a) is the original image, and (x.b)-(x.d) are reconstruction results of three images with different multi-distortion trade-off $\lambda_p$.

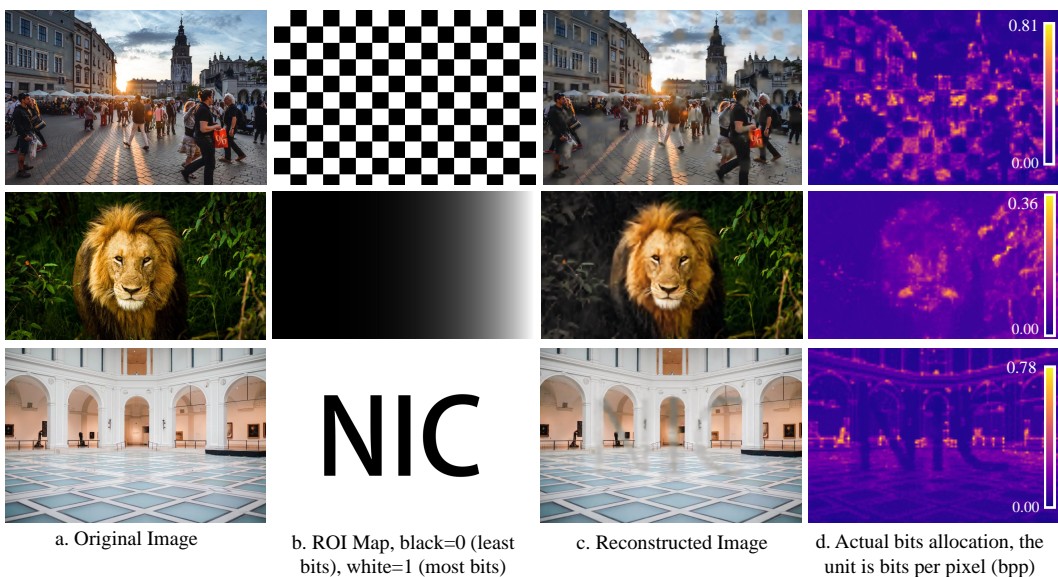

a. Original Image

b. ROI Map, black=0 (least bits), white=1 (most bits)

c. Reconstructed Image

d. Actual bits allocation, the unit is bits per pixel (bpp)

Figure 6: Results of Code Editing Enhanced's ROI-based coding.

Note that the major difference between our work and Yang et al. [2020] is that our approach requires only one decoder for continuous bitrate control, ROI and perception-distortion trade-off. And Yang et al. [2020] require multiple decoders for them. Yang et al. [2020] adopt SAVI Kim et al. [2018] to improve the R-D performance of a pair of encoder-decoder. We find SAVI can also be adopted to achieve bitrate control, ROI and multi-distortion with only a single decoder. In fact, even without the SGA of Yang et al. [2020], the semi-amortized inference of the simple AUN implementation can still achieve flexible bit rate control (See Sec. 4.2).

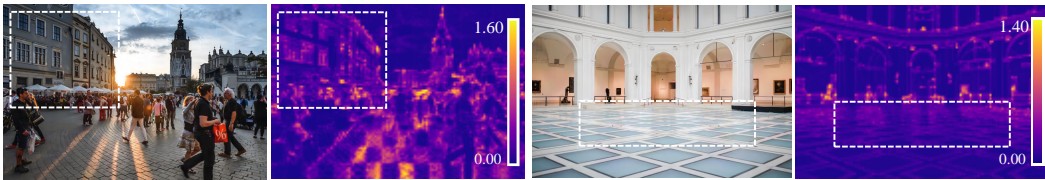

Figure 7: Results of Song et al. [2021]'s ROI-based Coding by using the same original image and ROI map of Fig. 6. The white rectangle indicates where ROI map is not effective.

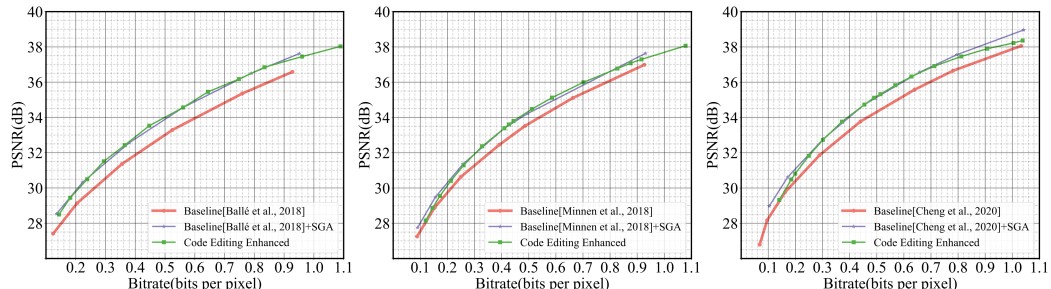

Figure 8: Comparison of R-D performance between Code Editing Enhanced and Baseline + SGA per model . Left: Ballé et al. [2018] as baseline. Middle: Minnen et al. [2018] as baseline. Right: Cheng et al. [2020] as baseline.

## 5    Conclusions

We propose Code Editing, a new paradigm for continuous variable bitrate NIC based on semi-amortized inference. It outperforms current variable bitrate NIC methods. Moreover, it achieves flexible spatial bits allocation and multi-distortion trade-off, all with one decoder. Limitations include that the efficiency of Code Editing could be improved, and that the empirical study on more compounded target mentioned as Eq. 10 should be addressed.

## Limitation

In general, our method does not work for the cases where encoding time matters, such as real-time communication. Our method is extremely useful for the cases where we encode just once but decode/view plural number of times, such as content delivery network. Another limitation of our work is that we limit the scope of discussion to scalar $\Delta$ instead of vector $\Delta$. Adopting a vector $\Delta$ increase the range of bitrate in ROI-based coding. For Code Editing Enhanced, all pixels share one $\Delta$. And this means in ROI-based coding, the overall $\Delta$ might not be suitable for all pixels with different quality map value $m$.

## Acknowledgments and Disclosure of Funding

This work is supported by SenseTime Research. The content is solely the responsibility of the authors and does not necessarily represent the official views of SenseTime Research.

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
