# Appendix

In Appendix A, extra quantitative results, including the results related to optimizing $\Delta$ and quantitative results on CLIC2020 [CLIC, 2022] are presented. In Appendix B, extra qualitative results are presented. And in Appendix C, we present the implementation details, including the detailed guidance to reproduce main results.

## A  More Quantitative Results

### A.1  Experimental Results on CLIC2022

The results in Sec. 4.3 are based on Kodak dataset. In this section we present the results comparing our Code Editing Enhance with other methods based on CLIC2022 dataset. From Fig . A1, we can see that the main conclusion remains unchanged.

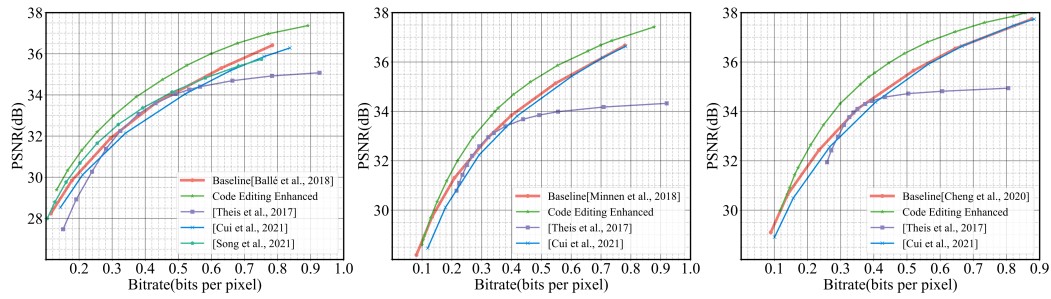

Figure A1: Comparison on CLIC [2022] dataset. Left: Ballé et al. [2018] as baseline. Middle: Minnen et al. [2018] as baseline. Right: Cheng et al. [2020] as baseline.

### A.2  Optimizing and Transferring $\Delta$

As shown in Fig. A2, using gradient method to optimize $\Delta$ of $z$ brings significant performance decay, while using gradient method to optimize $\Delta$ of $y$ is fine. So for Code Editing Enhanced, we use gradient based method to optimize $\Delta_y$, and grid search to optimize $\Delta_z$.

We set the grid search range as $\Delta_z \in \{2^{-1.5}, 2^{-1.0}, ..., 2^{1.5}\}$, and we pick $\Delta_z$ with the smallest R-D cost ($-\mathcal{L}_{y,z,\Delta_y,\Delta_z}$). The result of searching $\Delta_z$ is presented in Tab. A1. Note that as the value of $\Delta_z$ is discrete and has only 7 possibilities, we can simply use uniform prior, and we need only 3 bits

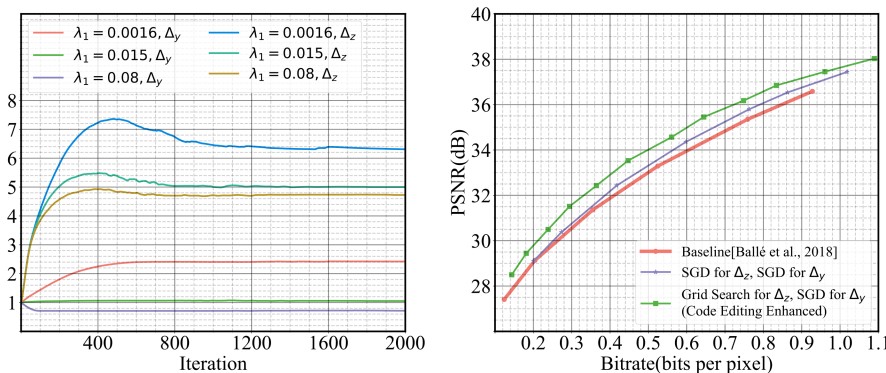

Figure A2: Comparison of gradient decent and grid search for optimizing $\Delta_z$. Left: the trend of $\Delta_y$ and $\Delta_z$ during gradient based optimization. Right: R-D performance comparison. SGD is the abbreviation of stochastic gradient descent, and in practical implementation we use Adam [Kingma and Ba, 2014] as optimizer.

Table A1: Results of grid search with $\lambda_0 = 0.015$ on Kodak dataset. $-\mathcal{L}_{y,\Delta}$ is shown in this table, lower is better.

| | $\lambda_1 = 0.0016$ | $\lambda_2 = 0.0032$ | $\lambda_3 = 0.0075$ | $\lambda_4 = 0.03$ | $\lambda_5 = 0.045$ | $\lambda_6 = 0.08$ |
|---|---|---|---|---|---|---|
| | Based on Ballé et al. [2018] | | | | | |
| $\Delta_z = 2^{-1.5}$ | 0.31686 | 0.44985 | 0.68303 | 1.24256 | 1.47532 | 1.89417 |
| $\Delta_z = 2^{-1.0}$ | 0.31419 | 0.44715 | 0.68061 | 1.24029 | 1.47309 | 1.89212 |
| $\Delta_z = 2^{-0.5}$ | 0.31168 | 0.44472 | 0.67832 | 1.23852 | 1.47096 | 1.89047 |
| $\Delta_z = 2^{0.0}$ | 0.30942 | 0.44253 | 0.67623 | **1.23741** | **1.47023** | **1.88992** |
| $\Delta_z = 2^{0.5}$ | 0.30788 | 0.44128 | **0.67568** | 1.23824 | 1.47200 | 1.89253 |
| $\Delta_z = 2^{1.0}$ | **0.30726** | **0.44126** | 0.67678 | 1.24219 | 1.47571 | 1.89552 |
| $\Delta_z = 2^{1.5}$ | 0.30895 | 0.44425 | 0.68222 | 1.25282 | 1.48909 | 1.91170 |
| | Based on Minnen et al. [2018] | | | | | |
| $\Delta_z = 2^{-1.5}$ | 0.32714 | 0.45260 | 0.66707 | 1.24483 | 1.49512 | 1.95127 |
| $\Delta_z = 2^{-1.0}$ | 0.32242 | 0.44801 | 0.66257 | 1.23849 | 1.49199 | 1.95180 |
| $\Delta_z = 2^{-0.5}$ | **0.31751** | **0.44376** | **0.65881** | 1.23064 | 1.48237 | 1.93762 |
| $\Delta_z = 2^{0.0}$ | 0.31902 | 0.44674 | 0.66350 | **1.23026** | **1.47921** | **1.93465** |
| $\Delta_z = 2^{0.5}$ | 0.32906 | 0.45899 | 0.67721 | 1.24036 | 1.48695 | 1.93773 |
| $\Delta_z = 2^{1.0}$ | 0.34360 | 0.47753 | 0.69713 | 1.25969 | 1.50642 | 1.95827 |
| $\Delta_z = 2^{1.5}$ | 0.34862 | 0.48562 | 0.70974 | 1.27931 | 1.52575 | 1.97856 |
| | Based on Cheng et al. [2020] | | | | | |
| $\Delta_z = 2^{-1.5}$ | 0.27705 | 0.39374 | 0.59356 | 1.12197 | 1.36487 | 1.84092 |
| $\Delta_z = 2^{-1.0}$ | 0.27571 | 0.39263 | 0.59231 | 1.12086 | 1.36374 | 1.83952 |
| $\Delta_z = 2^{-0.5}$ | 0.27465 | 0.39134 | 0.59127 | 1.11964 | 1.36280 | **1.83791** |
| $\Delta_z = 2^{0.0}$ | 0.27401 | 0.39034 | **0.59051** | **1.11934** | **1.36248** | 1.83809 |
| $\Delta_z = 2^{0.5}$ | **0.27309** | **0.39008** | 0.59065 | 1.12058 | 1.36370 | 1.83916 |
| $\Delta_z = 2^{1.0}$ | 0.27392 | 0.39136 | 0.59278 | 1.12506 | 1.36890 | 1.84496 |
| $\Delta_z = 2^{1.5}$ | 0.27644 | 0.39507 | 0.59910 | 1.13568 | 1.38064 | 1.85845 |

to encode it. And the value of $\Delta_y$ is a single-precision float which takes another 32 bits to encode. On Kodak dataset with $512 \times 768$ pixels, compressing both of the $\Delta$s requires around $8 \times 10^{-5}$ bpp. And on CLIC2022 which is even larger, the bpp of $\Delta$s is even less. So when presenting experimental results, we simply ignore the bpp of $\Delta$s.

Choi et al. [2019] also adopt the quantization stepsize control. However, our approach is very different from it. Specifically, we find there is train-test mismatch of the entropy model in Code Editing Naïve, which damages R-D performance. And then we propose adaptive quantization step to alleviate this problem (See Sec. 2.3 and Sec. 4.2). On the other hand, Choi et al. [2019] adjust the quantization step to fine-tune the bitrate. From the perspective of results, our proposed adaptive quantization works in the a wide bitrate region while the quantization step adjustment of Choi et al. [2019] works in a narrow bit rate region.

Moreover, Choi et al. [2019] sample $\Delta$ during training, which requires a carefully designed prior on $\Delta$. According to the original paper, training $\Delta \in [0.5, 2]$ brings best performance, and making it larger or narrower brings performance decay. While for us, the $\Delta$ is learned during SAVI stage and no deliberate prior is required. And during training we keep $\Delta = 1$ like a normal model. The advantage is that our method can be directly applied to any pre-trained neural compression model, while Choi et al. [2019] can not. Furthermore, we study the effect of optimizing $\Delta$ jointly with SAVI, which is never studied before. Moreover, we provide non-trivial extra insights into why this approach might work by theoretical analysis (Sec. 2.3) and empirical study (Sec. 4.2).

We note that it is also possible to treat $\Delta$s as a vector $\boldsymbol{\Delta}$ representing per dimension quantization step. In that case, the bpp of $\boldsymbol{\Delta}$ is unneglectable and has to be compressed separately. However, the vector $\boldsymbol{\Delta}$ is out of the scope of this paper.

### A.3  Go without Grid Search

Fig. A3 shows the result of fixing $\Delta_z = 1.0$ and performing no grid search at all. It can be seen that the R-D performance drop is only marginal compared with the grid search scheme in Sec. A.2 Therefore, it is possible to abandon the grid search and trade R-D performance for speed.

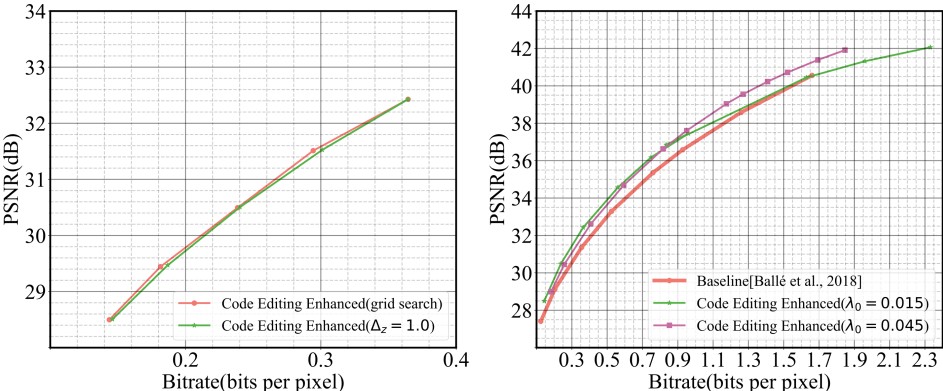

Figure A3: Left: The R-D performance between w/ and w/o grid search of $\Delta_z$. Right: The R-D performance beyond $bpp = 1.0$.

### A.4  Go Beyond $bpp = 1.0$

Fig. A3 shows the result of bpp beyond $1.0$. It can be observed that when we stick to a decoder trained with $bpp \approx 0.5$ ($\lambda = 0.015$), its R-D performance drops below the baseline when $bpp$ approaches 1.5. However, if we adopt a decoder trained with $bpp \approx 1.0$ ($\lambda = 0.045$), the R-D performance of our approach in high bitrate is significantly enhanced. We can achieve variable bitrate control without R-D performance loss in $0.1 \le bpp \le 1.9$.

## B  More Qualitative Results

### B.1  Multi-Distortion Trade-off

We present more qualitative results of multi-distortion trade-off in Fig. B1-B3. Both Kodak and CLIC2022 dataset are used.

### B.2  ROI-based Coding

We present more qualitative results of ROI-based coding in Fig. B4-B6. The ROI maps are shown in the main paper. Both Kodak and CLIC2022 dataset are used.

For segmentation based ROI, We selected the image 13e9b6 from the CLIC2022 dataset to test the segmentation ROI. There are 4 people in this image. We use separate segmentation for each person. Unlike the high contrast ROI shown in the main paper, we give the background a weight of 0.04 instead of 0. We define $PSNR_{region}$ to evaluate the image quality when ROI is used. Given an $m \times n$ image $I$ and the reconstructed image $K$, $MSE_{region}$ is defined as:

$$MSE_{\text{region}} = \frac{1}{\sum_{i=0}^{m-1}\sum_{j=0}^{n-1} M(i,j)} \sum_{i=0}^{m-1}\sum_{j=0}^{n-1} M(i,j)[I(i,j) - K(i,j)]^2$$

where $M(i,j)$ is 1(0) if pixel $(i,j)$ is inside(outside) the ROI region. $PSNR_{region}$ (in dB) is defined as:

$$PSNR_{\text{region}} = 10 \cdot \log_{10}\left(\frac{MAX_I^2}{MSE_{\text{region}}}\right)$$

These results can be found at Fig. B7-B9. We can see that code editing is effective for complex semantic ROI. And the visual quality of each person is improved accordingly as the ROI masks shift.

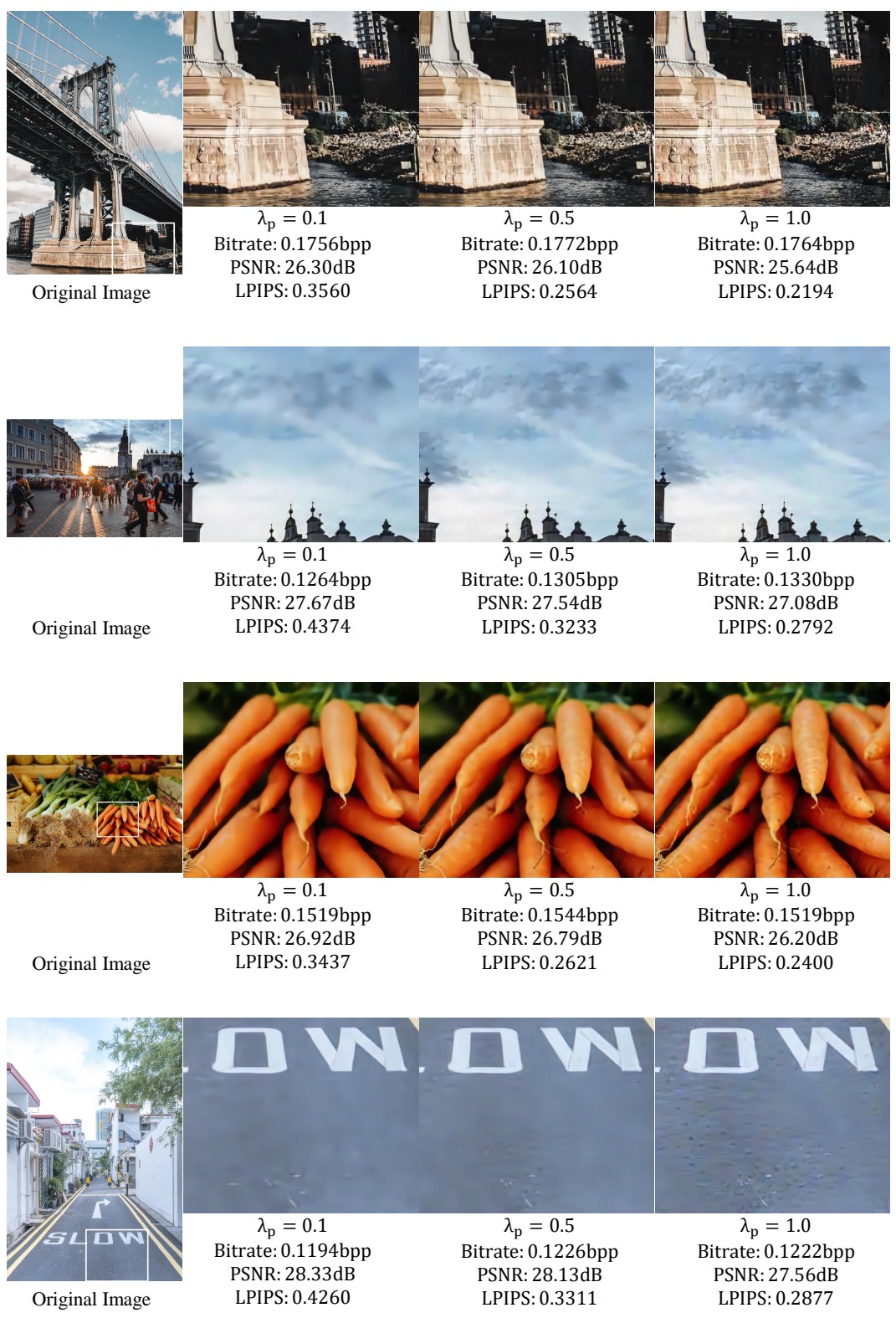

Figure B1: More multi-distortion trade-off results I.

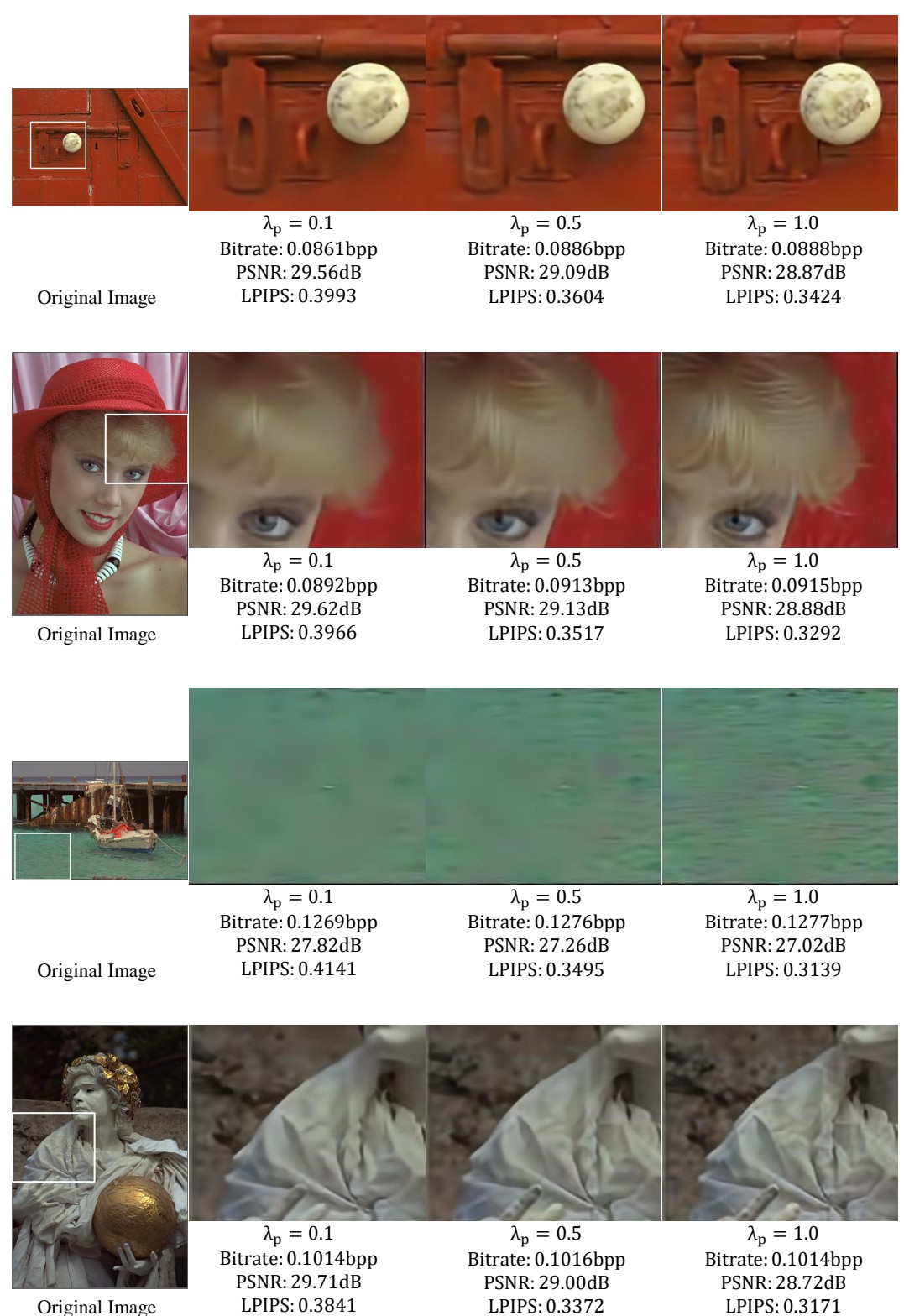

Figure B2: More multi-distortion trade-off results II.

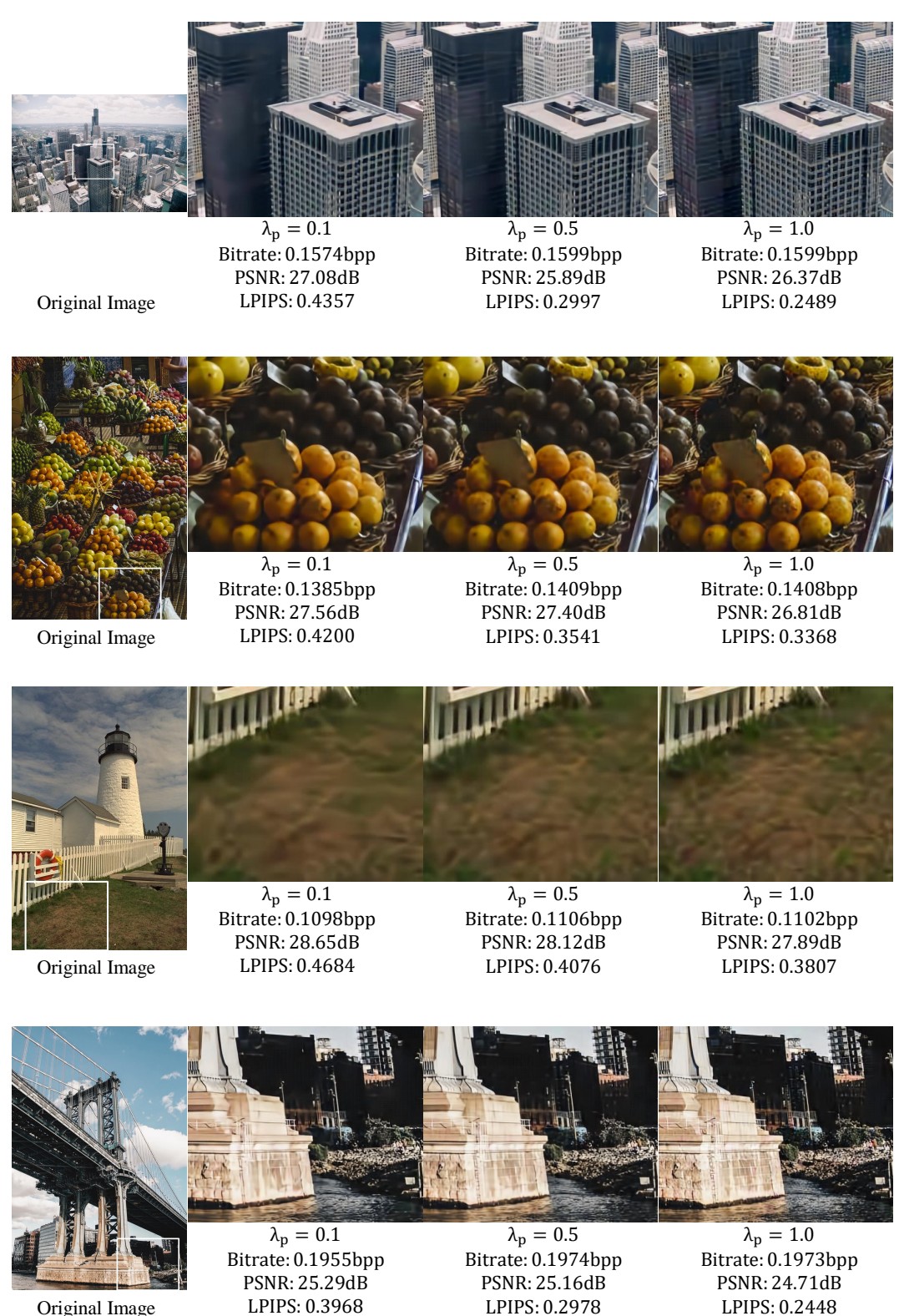

Original Image

$\lambda_p = 0.1$
Bitrate: 0.1574bpp
PSNR: 27.08dB
LPIPS: 0.4357

$\lambda_p = 0.5$
Bitrate: 0.1599bpp
PSNR: 25.89dB
LPIPS: 0.2997

$\lambda_p = 1.0$
Bitrate: 0.1599bpp
PSNR: 26.37dB
LPIPS: 0.2489

Original Image

$\lambda_p = 0.1$
Bitrate: 0.1385bpp
PSNR: 27.56dB
LPIPS: 0.4200

$\lambda_p = 0.5$
Bitrate: 0.1409bpp
PSNR: 27.40dB
LPIPS: 0.3541

$\lambda_p = 1.0$
Bitrate: 0.1408bpp
PSNR: 26.81dB
LPIPS: 0.3368

Original Image

$\lambda_p = 0.1$
Bitrate: 0.1098bpp
PSNR: 28.65dB
LPIPS: 0.4684

$\lambda_p = 0.5$
Bitrate: 0.1106bpp
PSNR: 28.12dB
LPIPS: 0.4076

$\lambda_p = 1.0$
Bitrate: 0.1102bpp
PSNR: 27.89dB
LPIPS: 0.3807

Original Image

$\lambda_p = 0.1$
Bitrate: 0.1955bpp
PSNR: 25.29dB
LPIPS: 0.3968

$\lambda_p = 0.5$
Bitrate: 0.1974bpp
PSNR: 25.16dB
LPIPS: 0.2978

$\lambda_p = 1.0$
Bitrate: 0.1973bpp
PSNR: 24.71dB
LPIPS: 0.2448

Figure B3: More multi-distortion trade-off results III.

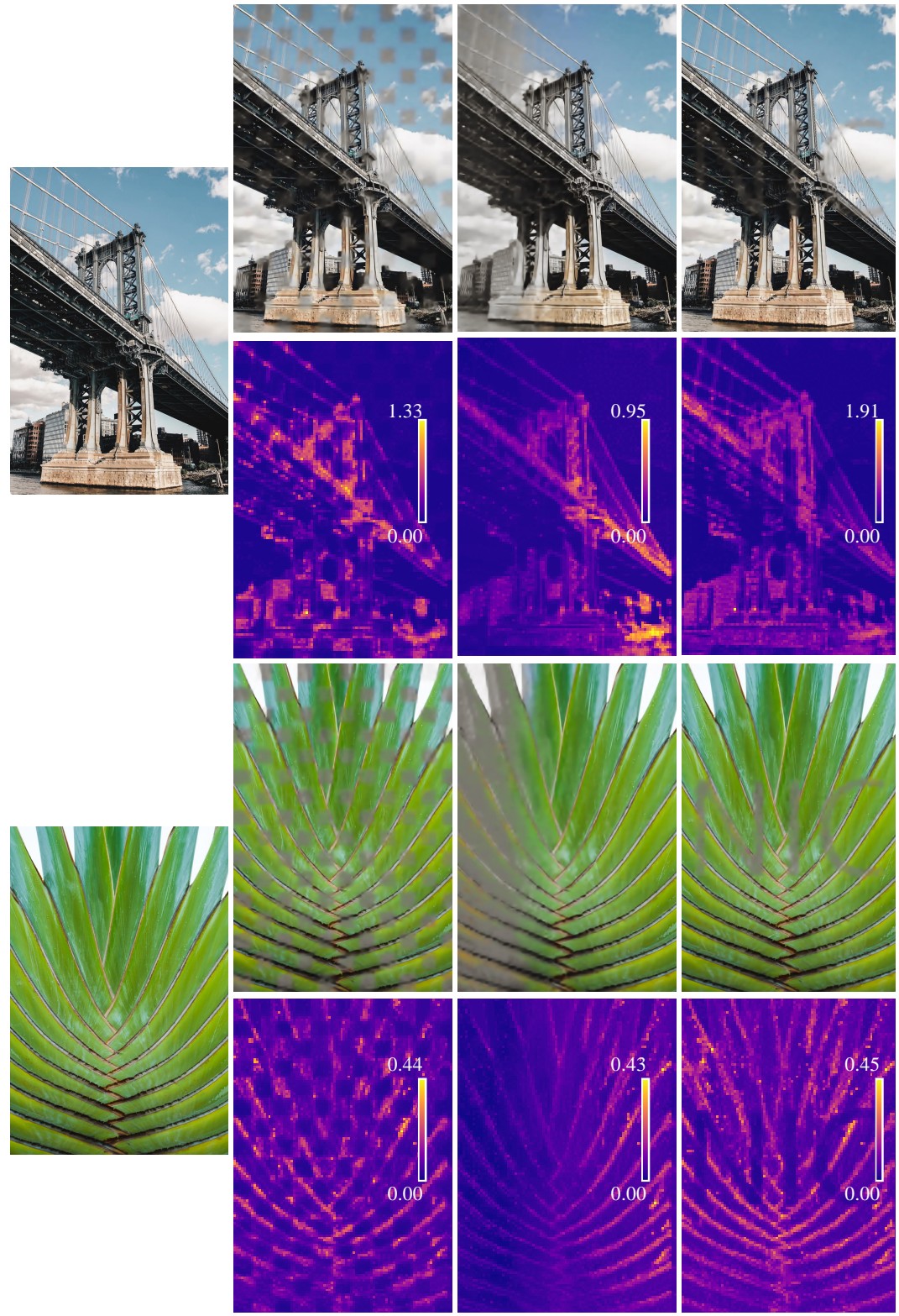

Figure B4: More ROI-based coding results I.

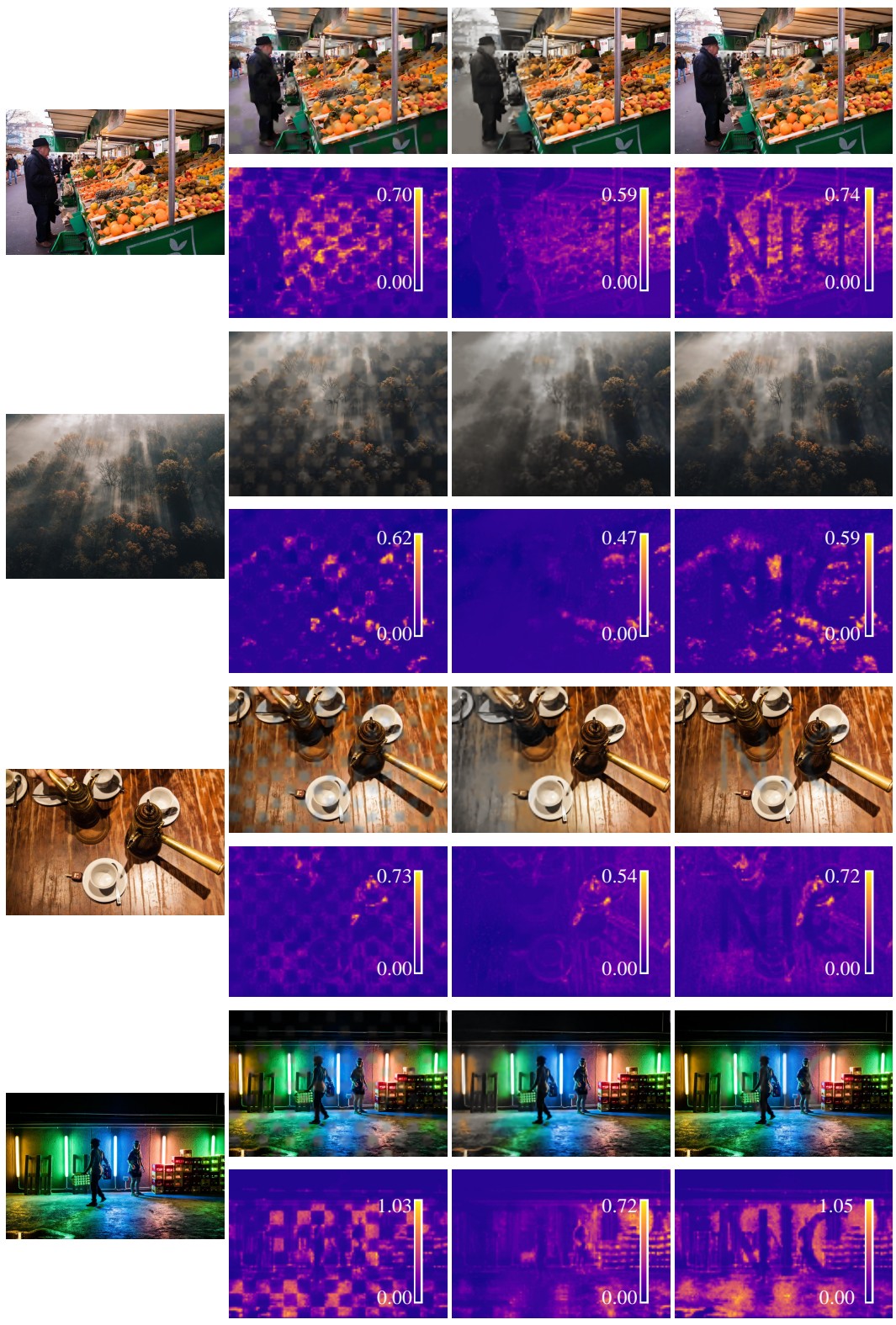

Figure B5: More ROI-based coding results II.

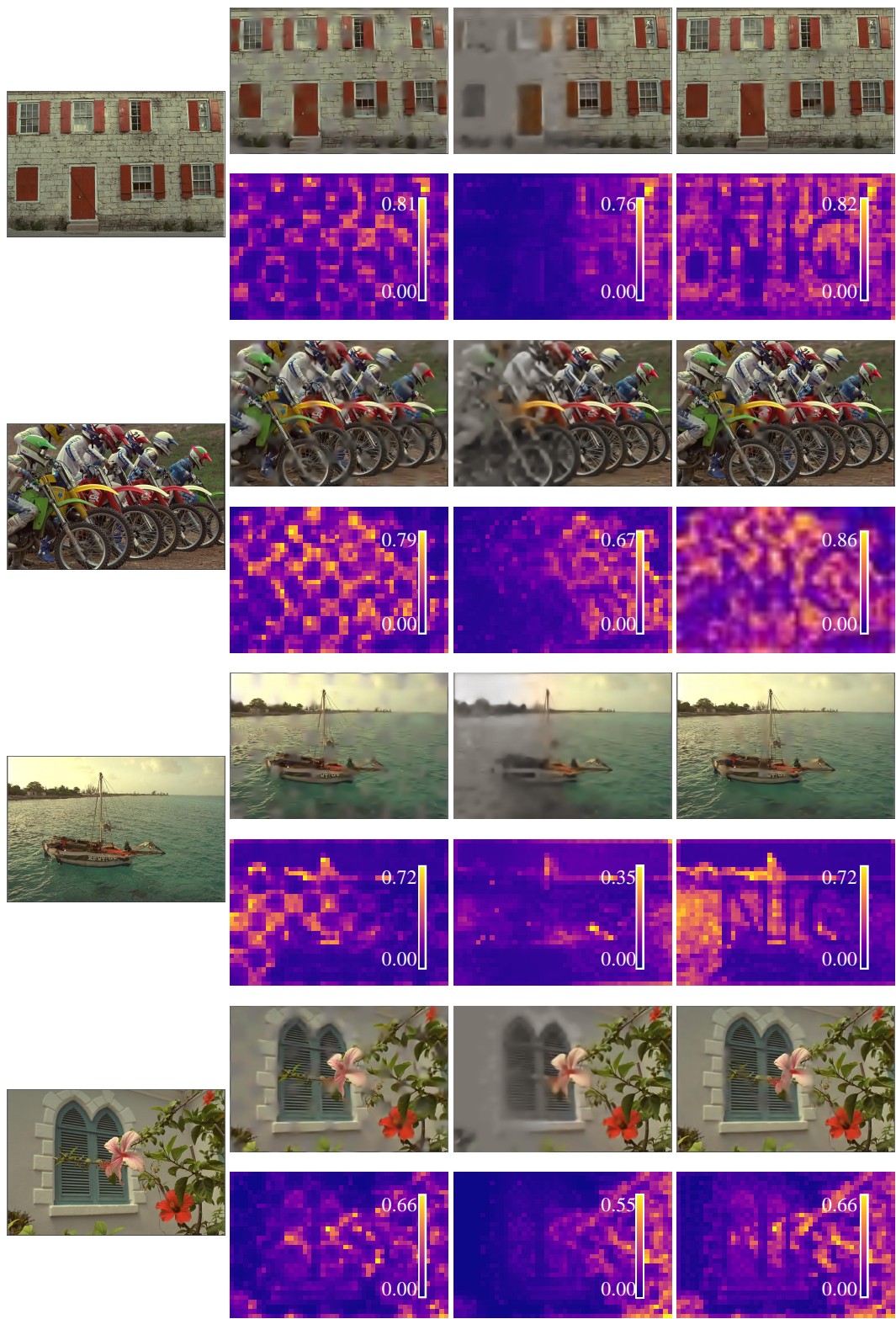

Figure B6: More ROI-based coding results III.

Original Image    ROI Map(baseline), black=0.04    Reconstructed Image

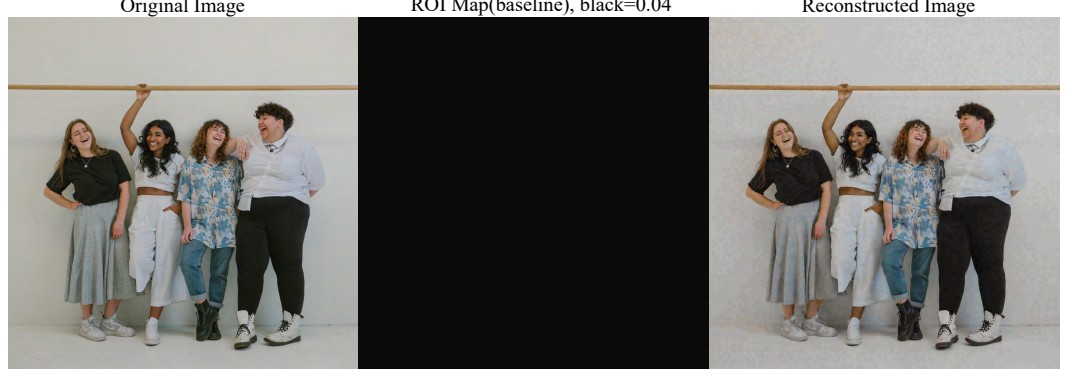

ROI Map(#1), black=0.04 (least bits), white=1 (most bits)    Reconstructed Image

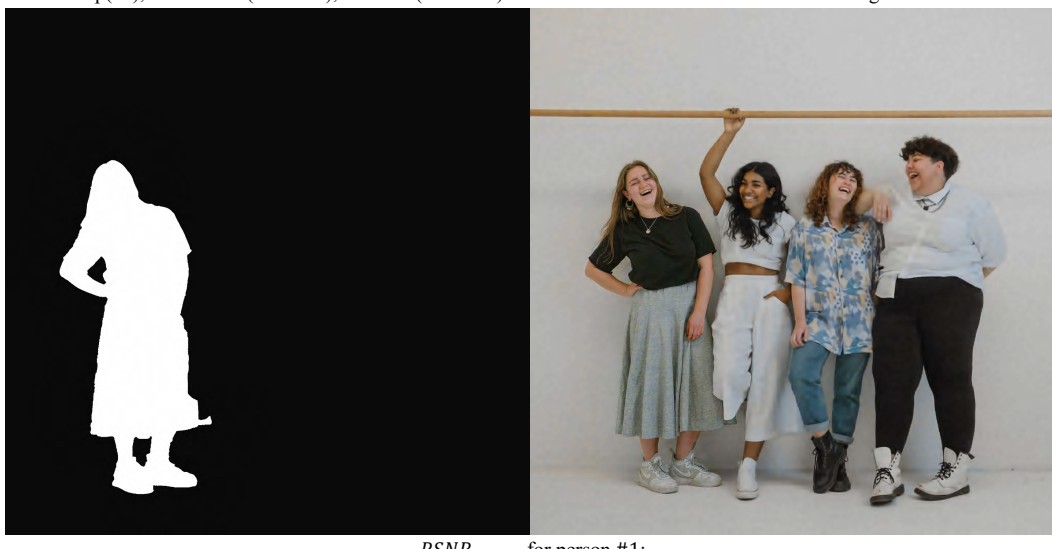

$PSNR_{region}$ for person #1:
(Baseline) → (#1)
$29.44dB → 33.17dB$

ROI Map(#2), black=0.04 (least bits), white=1 (most bits)    Reconstructed Image

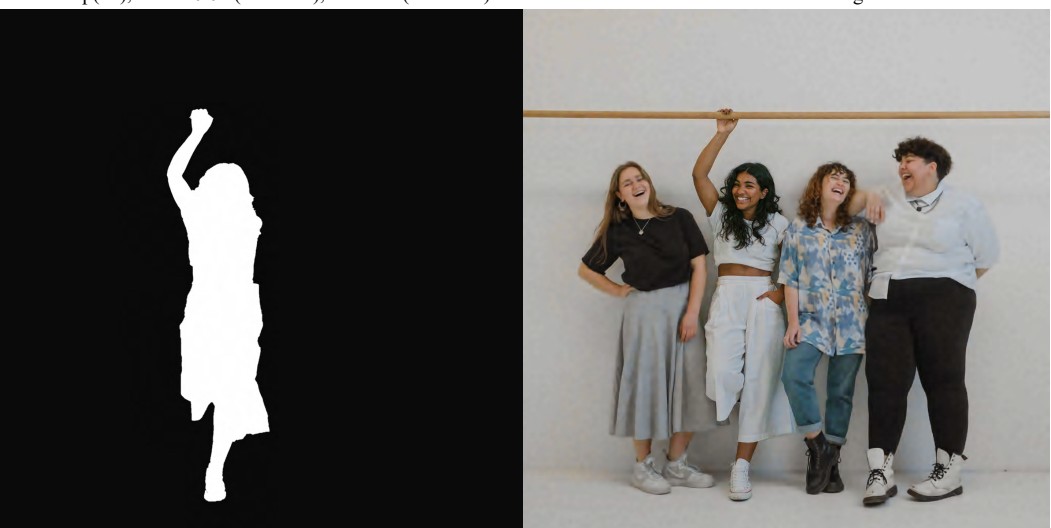

$PSNR_{region}$ for person #2:
(Baseline) → (#2)
$29.73dB → 34.47dB$

Figure B7: Segmentation based ROI coding results I.

ROI Map(#3), black=0.04 (least bits), white=1 (most bits)    Reconstructed Image

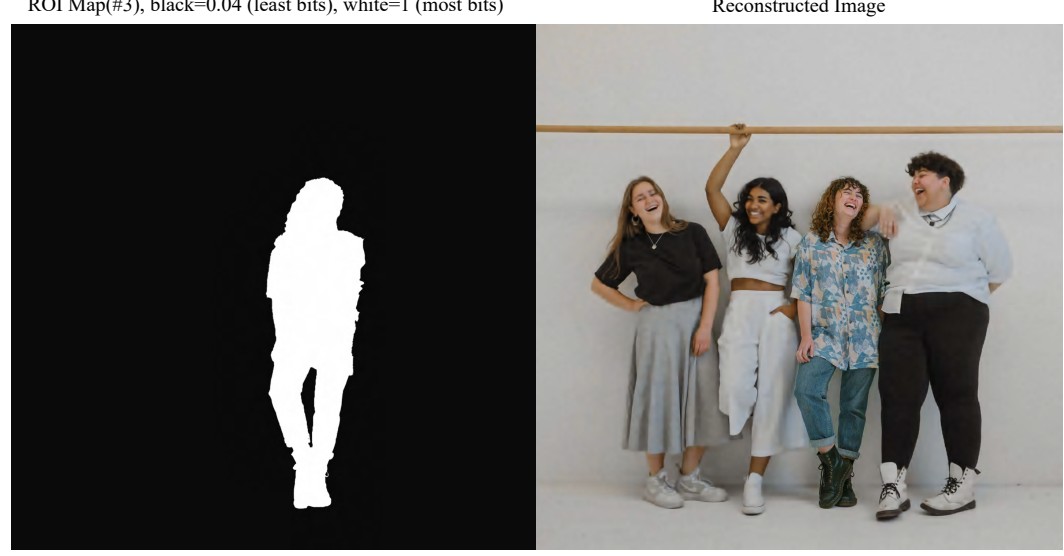

$PSNR_{region}$ for person #3:
(Baseline) → (#3)
$26.55dB → 32.57dB$

ROI Map(#4), black=0.04 (least bits), white=1 (most bits)    Reconstructed Image

$PSNR_{region}$ for person #4:
(Baseline) → (#4)
$30.99dB → 35.50dB$

Figure B8: Segmentation based ROI coding results II.

ROI Map(all people), black=0.04 (least bits), white=1 (most bits)          Reconstructed Image

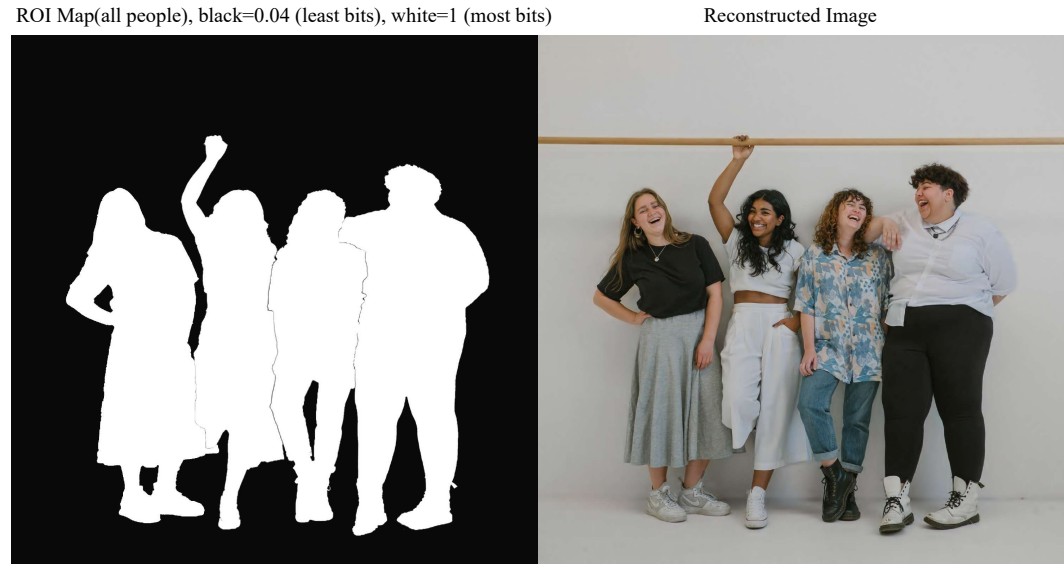

$PSNR_{region}$ for all people:
(Baseline) → (all people)
$29.08dB → 35.09dB$

Figure B9: Segmentation based ROI coding results III.

# C Experiment Details

## C.1 Detailed Experimental Settings

All the experiments are conducted on a computer with Intel(R) Xeon(R) CPU E5-2620 v4 @ 2.10GHz and $8\times$ Nvidia(R) TitanXp. All the experiments are implemented in Pytorch 1.7 and CUDA 9.0. The Hyper-parameters are aligned with Yang et al. [2020]. Specifically, we optimize $\Delta_y, y, z$ with SGA for $2,000$ iterations and learning rate $5 \times 10^{-3}$. More specifically for SGA, the temperature is set to $0.5 \times e^{-10^{-3}(k-700)}$, where $k$ is current epoch index. For Code Editing Enhanced, we search $\Delta_z$ to be $\{2^{1.5}, 2^{1.0}, 2^{0.5}, 2^{0.0}, 2^{-0.5}, 2^{-1.0}, 2^{-1.5}\}$.

## C.2 Guidance for Reproducing Main Results

**Baseline Models.** To reproduce the results in the main paper, we train the baseline models [Ballé et al., 2018, Minnen et al., 2018, Cheng et al., 2020]. We follow the settings of He et al. [2021]. These models are trained on the subset of 8,000 images from ImageNet for 2,000 epochs. We use Adam [Kingma and Ba, 2014] with learning rate of $10^{-4}$ and batch-size 16.

**Fig. 1 Left.** For Code Editing Enhanced, we use SGA[Yang et al., 2020] to optimize $y, z$. For Code Editing Enhanced(AUN), we use additive uniform noise as [Ballé et al., 2017, 2018, Cheng et al., 2020]. For Code Editing Naïve, we fix $\Delta_y$ and $\Delta_z$ to 1.0. For $\Delta_z$ optimization, we set $\Delta_z$ to $\{2^{1.5}, 2^{1.0}, 2^{0.5}, 2^{0.0}, 2^{-0.5}, 2^{-1.0}, 2^{-1.5}\}$.

**Fig. 1 Right.** We set iterations to 0, 50, 100, 150 and 200, respectively. Due to early termination, we adjust SGA temperature from original $0.5 \times e^{-10^{-3}(k-700)}$ to $0.5 \times e^{-10^{-3}(k-100)}$. For the red curves(w/o Encoder FT.), we use the $f_{\phi_{\lambda_0}}$ where $\lambda_0 = 0.015$. For green curves(w/ Encoder FT.), we use a finetuned encoder $f_{\phi_{\lambda'}}$. $f_{\phi_{\lambda'}}$ is finetuned from $f_{\phi_{\lambda_0}}$ on the training set for 500 epochs, with a learning rate of $10^{-4}$.

**Fig. 2.** We compute the histogram of normalized dequantized results of quantized $y^{(i)}$ before and after Code Editing on Kodak dataset which contains 24 images. And $i$ superscript indicates dimension. The source $\lambda_0$ is 0.015, and the target $\lambda_1$ is 0.0016. $y$ are obtained by $y \leftarrow f_{\phi_{\lambda_0}}(x)$. $y^*_{Na\"ive}$ and $y^*_{Enhanced}$ are obtained by Code Editing Naïve and Code Editing Enhanced, respectively. $\Delta^*$ is obtained by Code Editing Enhanced. All the dequantized results are normalized by $\sigma^2$, which is computed from $p(y|z)$ according to Ballé et al. [2018]. We get $11,796,480$ $y^{(i)}$s, and the bin size for the histogram is 1.

**Fig. 3.** For Song et al. [2021] and Cui et al. [2021], we add their proposed Spatial Feature Transform and Gain Unit to the the baseline models. We train these models using the same settings as training the baseline models. For Theis et al. [2017], we add the scale vectors to the baseline models trained under $\lambda_0 = 0.015$ and finetune the scale vectors for 100 epochs, with a learning rate of $10^{-4}$.

**Fig. 4.** These curves are obtained by optimizing $y$ and $\Delta$ to maximize $\mathcal{L}^{MD}_{y,\Delta}$(see Eq. 9). We adopt a bitrate control scheme from Mentzer et al. [2020] to make the bitrate roughly same. First, we set $R_{target}$ to the bitrate before editing for each image. Then, we add a weight $\lambda_r$ to the left term $\log P_{\theta_{\lambda_0}}(\lfloor y/\Delta \rceil; \Delta)$ in Eq. 9. $\lambda_r$ is set to $\lambda^{(a)}$ and $\lambda^{(b)}$ for $-\log P_{\theta_{\lambda_0}}(\lfloor y/\Delta \rceil; \Delta) > R_{target}$ and $-\log P_{\theta_{\lambda_0}}(\lfloor y/\Delta \rceil; \Delta) < R_{target}$, respectively, where $\lambda^{(a)} \gg \lambda^{(b)}$. Here, we set $\lambda^{(a)}$ to 4.0 and $\lambda^{(b)}$ to 0.25.

**Fig. 5.** The setting is same as **Fig.4**. These two images are from CLIC2022[CLIC, 2022] dataset, where IDs are 5ab399 and 6f2e3f. The cropping positions $(left, upper, right, lower)$ are $(385, 1519, 865, 1919)$ and $(0, 1300, 480, 1700)$, respectively.

**Fig. 6.** We optimize the target $\mathcal{L}^{ROI}_{y,\Delta}$ to obtain the ROI-based coding results. Here, we adopt the baseline model Ballé et al. [2018] trained under $\lambda_0 = 0.0016$. We conduct the experiment on Kodak and CLIC2022 datasets. We sample 3 images and show them in **Fig. 6**. They are from CLIC2022 dataset. Their IDs are 76e721, 2ff706 and 631791, respectively.

**Fig. 7.** For a fair comparison, we adapt the framework proposed by Song et al. [2021] to the baseline model Ballé et al. [2018]. We feed the quality map together with the image to this model to get the ROI-based coding results of Song et al. [2021].

## Broader Impact

In CLIC (Challenge on Learned Image Compression) 2021, the champion solution Gao et al. [2021]'s decoder size is 230MB. On the other hand, Savinaud et al. [2013] as a JPEG2000 codec is only around 1.4MB. Thus, a flexible neural codec with one decoder for all conditions is highly valuable in terms of disk storage reduction, which in turn benefits carbon footprint reduction. On the other hand, our solution enables continuous control of bitrate with flexible ROI. This makes NIC as flexible as traditional codecs, and thus prompts the practical deployment of NIC. Moreover, the distortion-perception trade-off also attracts attentions of contemporary traditional codecs, such as H.266 [Bross et al., 2021].