# OpenReview forum: "Flexible Neural Image Compression via Code Editing"
_NeurIPS.cc/2022/Conference — NeurIPS 2022 Accept_

### Official Review · Reviewer_JZkG · 2022-07-06

**Rating:** 8
**Confidence:** 5
**Soundness:** 4 excellent
**Presentation:** 3 good
**Contribution:** 4 excellent

**Summary:**

This paper proposes a method called "Code Editing" for rate control, which is an optimization of the latents at encode time. This has commonly been used in the past to boost rate distortion performance, but not as a continuous rate control mechanism. Second, this paper proposes a method of combating the rate-distortion decay of latent optimization with the use of an adaptive quantizations step size. Last, the paper allows for a per pixel spatial rate-allocation mechanism that grants even more fine-grained control of the rate.

Given the authors responses, I have raised my overall review.

**Questions:**

1. What is the rate distortion performance at very high rates (>1.0 bpp, say 2.0)? Is code editing still effective at high rates?
2. Is code editing effective with ROI across complex ROI masks? e.g. imagine you have an image with 3 faces roughly the same size, can shifting the ROI to each of the three faces increase fidelity equally?
3. How important is the granularity of the quantization step size being searched? If the grid were two or four times as dense, would you have increased performance or just more points on the RD curve?
4. Many of the baseline models have increased capacity for the larger rates in order to keep quality high. For example, in the Balle 2018 case, was the model using the 192 or 320 throughout? Are higher capacity models needed to allow for the flexibility needed for code editing?

**Limitations:**

Author sufficiently discussed the limitations of their work and any potential negative society impact.

**Strengths And Weaknesses:**

Strengths:
Through "code editing", the latents can be additionally optimized for the given rate-distortion trade-off over a large range of bitrates [0.1 to 1.0] bpp. This allows one model, with additional encoding time to target a wide swatch of the rate-distortion curve.

Per pixel rate control allows a very explicit, fine grained rate control on a per image basis. Additionally, this information is only used at encode time and isn't required to be transmitted to the decoder.

The empirical results of naive code editing only being successful in a narrow rate range (around 0.1 bpp).

Weaknesses:
No results were shown in the very high quality/high rate range. From the rate-distortion curves, it appears as if the Code Editing method starts to flatten at the high rate (see fig 3), versus the ideal range of around 0.5 bpp.

The ROI based coding examples seem very much designed to show high contrast (checkerboard, gradient, NIC). Show the result of a semantic segmentation model's ROI and showing if bit allocation on small text or faces can enhance the overall image at low bitrates would make an obvious visual difference. (Perhaps the ROI shifts between one of several faces in the same image for example).

The quantization step size is currently determined via grid search (brute force).

---

> ### Author Response · Authors · 2022-08-01
> **Response to the comments of Reviewer JZkG**
>
> Thanks for your detailed review. And we are glad to provide our answer to your questions:
>
> ### Q1 & W1 Results in the very high quality/high rate range
> * With [Ballé et al. 2018] as the baseline, we tested Code Editing Enhanced at a much longer bitrate range (up to 2.3 bpp) and these results are available at [JZkG-1](https://anonymous.4open.science/api/repo/NeurIPS268_Materials/file/JZkG-1.pdf). It can be seen that Code Editing Enhanced does start to flatten for higher bit rates and cannot outperform the baseline at very high rate (>1.5 bpp). We hypothesis the bottleneck of the base model trained at a relatively low bitrate (around 0.5 bpp) limits R-D performance in the very high bitrate range. So we tested Code Editing Enhanced at another base model where the R-D trade-off parameter $\lambda_0$ is set to 0.045 (around 1.0 bpp). We can see that after changing the base model's R-D trade-off to a higher bitrate, the flattened R-D curve has been lifted again.
> * __For revision__: We have included the discussions.
>
> ### Q2 & W2 Segmentation ROI
> * Thanks for your advice. We selected the image 13e9b6 from the CLIC2022 [CLIC, 2022] test set to test the segmentation ROI. There are 4 people in this image. We use separate segmentation for each person. Unlike the high contrast ROI shown in the main paper, we give the background a weight of 0.04 instead of 0. These results can be found at [JZkG-2](https://anonymous.4open.science/api/repo/NeurIPS268_Materials/file/JZkG-2.pdf) ([Ballé et al., 2018] as the base model, $\lambda_0=0.015$). We can see that code editing is effective for complex semantic ROI. The visual quality and fidelity measured by PSNR of each person are improved accordingly as the ROI masks shift.
> * __For revision__: We have included those results.
>
> ### Q3 & W3 grid search for quantization step size
> * As we discussed in Appendix A.3, grid search is only used to optimize $\Delta_z$, and $\Delta_y$ is optimized by gradient descent. As the bitrate of $z$ is only a small part of the encoding (less than 10\%), so the impact of $\Delta_z$ is marginal. In fact, even the current grid search framework for $\Delta_z$ is a bit of unnecessary. In the extreme case, we can choose to fix $\Delta_z=1$ and only optimize $\Delta_y$. This result is shown in [JZkG-3](https://anonymous.4open.science/api/repo/NeurIPS268_Materials/file/JZkG-3.pdf) ([Ballé et al., 2018] as the base model, $\lambda_0=0.015$). We can see that without grid search at all, the performance of our approach is only marginally effected.
> * __For revision__: We have clarified that grid search is only used to optimize $\Delta_z$ in Sec. 2.6.
>
> ### Q4 Many of the baseline models have increased capacity for the larger rates in order to keep quality high. For example, in the Balle 2018 case, was the model using the 192 or 320 throughout? Are higher capacity models needed to allow for the flexibility needed for code editing?
> * For all the experiments in the paper, we follow the exact setting of the baselines. This means that for [Ballé et al., 2018] we use the model with $320$ channels for all range as the original paper uses $320$ channels model to train $\lambda=0.015$.
> * The impact of model capacity is indeed an interesting topic, while we are not sure about how to include this in our experiment. As our major claim is to achieve variable bitrate with single decoder, changing channel capacity in the middle does not look like an option. However, using channel size $192$ for high bitrate range degrades the performance for sure as it also degrades the performance for the baseline [Ballé et al., 2018].
> * Without changing the model capacity, we can improve the R-D performance in the high quality range by changing the R-D trade-off of the base model. This is dicussed in Q1 & W1 and relative results are shown in [JZkG-1](https://anonymous.4open.science/api/repo/NeurIPS268_Materials/file/JZkG-1.pdf).
>
> ### Reference
> * J. Ballé, D. Minnen, S. Singh, S. J. Hwang, and N. Johnston. Variational image compression with a scale hyperprior. In International Conference on Learning Representations, 2018.
> * CLIC. Workshop and challenge on learned image compression (clic). http://clic.compression.cc/, 2022.

---

### Official Review · Reviewer_kNRz · 2022-07-09

**Rating:** 6
**Confidence:** 4
**Soundness:** 1 poor
**Presentation:** 3 good
**Contribution:** 3 good

**Summary:**

The paper investigates obtaining flexible-rate neural image compression (NIC) methods by adapting representations during encoding. They obtain a new representation y' by adapting the quantization width as well as the latent values themselves. Parameters are frozen so no overhead except for longer encoding time.

**Questions:**

Q1) Did you experiment with adapting $\Delta$ during training also? I found it surprising that you can use a model trained for lambda  = 0.015

**Limitations:**

No negative societal impact described in the main text, hidden in the supplementary material.

**Strengths And Weaknesses:**

Overall solid idea. I'm not super familiar with the adaptive rate literature but it seems novel to me. I like the idea of adapting $\Delta$. I also appreciate that there are various interesting ablation studies in Fig 1, and that authors investigate using the formulation for perceptual quality optimization and adaptive ROI-based rates (neat!). The results in Fig. 3 are promising.

## Weaknesses

- W1) Bad formulation in Sec 2.1: To me, who is very familiar with this field, Sec 2.1, in particular L72-L84 sounded weird. What is the meaning of "connected". What is it supposed to mean that the "factorized uniform distribution is used to simulate the quantization noise" (I suppose that we add IID uniform random noise?). What about "the decoding process is connected to compute the likelihood ..., which is equivalent to distortion evaluation" (some link to VAE?). This might be a language barrier issue but it sounds to me like some badly pieced together bits from Balle et al's original paper. I would strongly suggest rewriting Sec 2.1.
- W2) Incorrect/bad formulation in L112: "In NIC, the probability mass function (pmf) [...] over quantized y is computed via differentiating cumulative distribution function" [sic]. That is only true for the factorized prior from Balle et al, which is formulated via a CDF. For e.g. hyperprior based approaches (used in almost every compression paper since), we parameterize the density directly and calculate the PMF by integrating over boxes.
- W3) The above makes me question the soundness of the evaluation. Did the authors use range coding/arithmetic coding in the end to calculate real bitrates? Was care taken to make sure the model does not accidentally cheat?

- W4) (minor) similar to the above, the recap of distortion perception left a bad taste in my mouth. The authors cite Blau and Michaeli's work, where the perceptual loss is formulated via a _divergence_, yet authors use LPIPS, a normal pair-wise distortion, to do perceptual optimization.

## Conclusion

Overall, I will have to give a Reject rating, but if authors can convince me that they i) did a solid evaluation including range coding that implies we can trust the results and that they ii) will make sure the formulation is sound and understandable, I am willing to change my mind, since the approach itself looks interesting.

I'm giving a 4/5 confidence rating since there might be related work to adaptive-rate NIC that I'm missing

---

> ### Author Response · Authors · 2022-07-28
> **Response to the comments of Reviewer kNRz Part I**
>
> Thanks for your detailed review. And we are glad to provide our answer to your questions, and a few clarification on some misunderstandings:
>
> ### W1 Bad formulation in Sec 2.1:
>
> * The formulation presented in L72-L84 (before revision) comes from variational autoencoder (VAE) [Kingma and Welling 2013], which is widely adopted in NIC works such as [Ballé et al., 2018, Yang et al. 2020]. Specifically, the encoder in compression corresponds to the inference model in VAE, the decoder in compression corresponds to the generative model in VAE.
> * The word “connected” means that the iid additive uniform noise used to relax discrete latent is equivalent to the reparameterization of variational posterior $q(\tilde{y}|x)$. The "factorized uniform distribution is used to simulate the quantization noise" means that we add iid uniform random noise just as you supposed. We emphasis “factorize” and “uniform” here as the variational posterior $q(\tilde{y}|x)$ is factorized uniform distribution.
> * The data likelihood term $\log p(x|\tilde{y})$ is known to be related to the distortion metric $d(x,\bar{x})$. For example, when the distortion metric is MSE and data likelihood is factorized Gaussian, we can set the decoder’s output $\bar{x}$ be the $\mu$ and $\sigma^2=1/2\lambda$, then the data likelihood term $\log p(x|\tilde{y})$ in ELBO is just the $\lambda * MSE + constant$. This is where the equivalence comes from. In fact, [Minnen et al., 2018] even extent this connection beyond Gaussian and MSE. The general idea is that if we treat the distortion $d(.,.)$ as the energy function, then the likelihood term is equivalent to the likelihood of Gibbs distribution defined by such energy function.
>
> * __For revision__: we have rewritten this section to make it clearer.
>
> ### W2 Incorrect/bad formulation in L112 (before revision)
>
> * We think the phrase "differentiating cdf" is abused in L112 (before revision). We want to express taking the difference of cdf by "differentiating cdf", instead of taking the gradient. For factorized Gaussian $p(\tilde{y}|\tilde{z})$ in hyperprior based approaches, integrating the pdf and taking the difference of the cdf produces the same result. As a matter of fact, denote the pdf of random variable as $p(x)$, and the cdf as $F(x)$, we have $P(x_1<x\le x_2)=\int_{x_1}^{x_2}p(x)dx=F(x)|_{x_1}^{x_2}=F(x_2)-F(x_1)$. In practice, we can also implement this using cdf as $dist.cdf(y+0.5)-dist.cdf(y-0.5)$.
> * __For revision__: we have revised the expression here.
>
> ### W3 soundness of the evaluation
> * For all the results reported in the R-D curve, the bitrate is measured by actual bits of range encoder, and the reconstruction is computed by the latent coded from the actual the range decoder. For all visualization involves spatial bitrate distribution, the theoretical bitrate is used. We base our paper on a mature internal library for neural compression and we did not implement the range codec by ourselves. That is the reason why we did not mention it in paper as we took it for granted. To clarify, we will add a section to emphasis this.
>
> * __For revision__: we have added a section to emphasis this.
>
> ### W4 (minor) distortion perception
> * We agree. In fact the original plan is to use GAN loss but we end up with LPIPS [Zhang et al., 2018]. We will rename the distortion-perception trade-off into multiple-distortion trade-off. We fully admire [Blau and Michaeli, 2018, 2019] and we do not want to use the term "distortion-perception trade-off" with LPIPS.
> * __For revision__: we have renamed distortion-perception trade-off into multiple-distortion trade-off.

---

> > ### Comment · Reviewer_kNRz · 2022-08-05
> > **Response**
> >
> > Thanks for the response. The reply convinces me that the authors have sound evaluation and will be able to improve the formulation. I will change my rating to accept.

---

> ### Author Response · Authors · 2022-07-28
> **Response to the comments of Reviewer kNRz Part II**
>
> ### Q1 $\Delta$ during training
> * We fix $\Delta=1$ during training, which is different from [Theis et al. 2017] and [Choi et al. 2019]. Training with different $\Delta$ requires careful design of prior on $\Delta$, and ties our method to encoder-decoder trained with specific approach. According to [Choi et al. 2019], training $\Delta \in [0.5,2]$ brings best performance, and making it larger or narrower bring performance decay. The major advantages of our approach are: (1) it is prior free, (2). it can be directly applied to any pre-trained neural image compression model. Training with $\Delta$ ruins those features.
> * As our distortion is measured in $0-255$ instead of $0-1$, the $\lambda=0.015$ is around the middle bpp (0.5). It is not surprising that a decoder trained with middle bpp can decode the image with low and high bpp. Despite The empirical result is very promising, it is authentic as the reported R-D performance has been through actual entropy encoding and decoding.
>
> ### Q2 No negative societal impact described in the main text, hidden in the supplementary material.
> * __For revision__: we will make some space for this section in the main text.
>
> ### Reference
> * J. Ballé, D. Minnen, S. Singh, S. J. Hwang, and N. Johnston. Variational image compression with a scale hyperprior. In International Conference on Learning Representations, 2018.
> * D. Minnen, J. Ballé, and G. D. Toderici. Joint autoregressive and hierarchical priors for learned image compression. Advances in neural information processing systems, 31, 2018
> * R. Zhang, P. Isola, A. A. Efros, E. Shechtman, and O. Wang. The unreasonable effectiveness of deep features as a perceptual metric. In CVPR, 2018
> * Y. Blau and T. Michaeli. The perception-distortion tradeoff. In Proceedings of the IEEE conference on computer vision and pattern recognition, pages 6228–6237, 2018.
> * Y. Blau and T. Michaeli. Rethinking lossy compression: The rate-distortion-perception tradeoff. In International Conference on Machine Learning, pages 675–685. PMLR, 2019.
> * Y. Yang, R. Bamler, and S. Mandt. Improving inference for neural image compression. Advances in Neural Information Processing Systems, 33:573–584, 2020.

---

### Official Review · Reviewer_Q7gH · 2022-07-09

**Rating:** 6
**Confidence:** 3
**Soundness:** 3 good
**Presentation:** 2 fair
**Contribution:** 2 fair

**Summary:**

- The authors propose Code Editing that control bitrate of neural image compression with semi-amortized inference.
- The authors solved the performance decay in low rate of Code Editing by making the quantization step size adaptive.

**Questions:**

- When comparing the R-D curves of Code Editing Enhanced and Yang et al. [2020], what is the difference of the proposed method and is it reasonable?
- Why does combining semi-amortized inference and adaptive quantization step size improve when low-rate results?
- Why are the low-rate results in Fig. 3 approaching baseline?

**Limitations:**

- In Chapter 5, the authors told that efficiency of Code Editing could be improved with regard to limitations, but it would be good to be more specific about what cases the efficiency should be improved. For example, what are the cases in which the authors' method does not work well?

**Strengths And Weaknesses:**

Strengths
- The authors shows an interesting experimental result (Fig. 1 Left) it seems to solve the performance decay in low rate of Code Editing.

Weakness
- The new part of method has not seem to be clearly explained. The basic parts of amortized inference strategy in neural image compression has already been proposed (Yang et al. [2020]). Therefore, I thought it was a bit exaggerated when section 2.2 states that they propose a new paradigm of controlling R-D trade-off by semi-amortized inference. Variable bitrate compression by changing the quantization step size has also been proposed for neural image compression (Choi et al. [2019]). It would be great if you could clarify what is new when compared to these previous studies.
- The R-D curve of Code Editing Enhanced versus conventional semi-amortized inference based neural image compression (Yang et al. [2020]) has not been compared. If quantitative comparisons with previous studies were made (e.g. overhead to support variable bitrate), it would be easier to argue the superiority of this study.
- typo: p.2 L53 bitrtae -> bitrate

---

> ### Author Response · Authors · 2022-07-29
> **Response to the comments of Reviewer Q7gH Part I**
>
> Thanks for your detailed review. And we are glad to provide our answer to your questions, and a few clarification on some misunderstandings:
>
> ### W1 The new part of method has not seem to be clearly explained
> * The major difference between our work and [Yang et al. 2020] is that our approach
> requires only one decoder for continuous bitrate control, ROI and perception-distortion (multiple-distortion) trade-off. And [Yang et al. 2020] require multiple decoders for them. [Yang et al. 2020] adopt SAVI [Kim et al., 2018] to improve the R-D performance of a pair of encoder-decoder. We find SAVI can also be adopted to achieve bitrate control, ROI and perception-distortion (multiple-distortion) with only a single decoder. In fact, even without the SGA of  [Yang et al. 2020], the semi-amortized inference of the simple AUN implementation can still achieve flexible bit rate control (Sec 4.2--SGA vs. AUN).  Our main contribution is to explore flexible NIC with semi-amortized inference, instead improving R-D performance.
> * We disagree that the the claim of a new paradigm of controlling R-D trade-off is exaggerated. As [Yang et al. 2020] only use SAVI to improve R-D performance, it does not consider controlling R-D trade-off. And we are indeed the first to adopt SAVI for controlling R-D trade-off. Thus, it is a new paradigm of controlling R-D trade-off.
> * We agree that changing the quantization step size has been proposed and we have properly cited [Choi et al. 2019]. However, our approach is very different from previous ones. Specifically, we find there is train-test mismatch of the entropy model in Code Editing Naïve, which damages R-D performance. And then we propose adaptive quantization step to alleviate this problem (Sec 2.3 and Sec 4.2). On the other hand, [Choi et al. 2019] adjust the quantization step to fine-tune the bitrate. From the perspective of results, our proposed adaptive quantization works in the a wide bitrate region while the quantization step adjustment of [Choi et al. 2019] works in a narrow bit rate region.
> * Moreover, [Choi et al. 2019] sample $\Delta$ during training, which requires a carefully designed prior on $\Delta$. According to the original paper, training $\Delta \in [0.5,2]$ brings best performance, and making it larger or narrower brings performance decay. While for us, the $\Delta$ is learned during SAVI stage and no deliberate prior is required. And during training we keep $\Delta=1$ like a normal model. The advantage is that our method can be directly applied to any pre-trained neural compression model, while [Choi et al. 2019] can not. Furthermore, we study the effect of optimizing $\Delta$ jointly with SAVI, which is never studied before. Moreover, we provide non-trivial extra insights into why this approach might work by theoretical analysis (Sec 2.3) and empirical study (4.2 and Fig. 2).
> * __For revision__: we have included the discussions.
> ### W2 The R-D curve of Code Editing Enhanced vs [Yang et al. 2020]
> * It has been compared in Appendix A.1 Fig A.1. We will move it to main text to make it more obvious. We are neither superior nor inferior to [Yang et al. 2020], since we are considering vastly different tasks. [Yang et al. 2020] aim at improving R-D performance, and we aim at continuously control bitrate with one decoder. We can not compare with [Yang et al. 2020] in terms of the overhead to support variable bitrate, as [Yang et al. 2020] does not support variable bitrate.
> * __For revision__: we will make some space and move this experiment to main text.
> ### W3 typo: p.2 L53 bitrtae -> bitrate
> * Thanks for pointing it out, we will fix it.
> * __For revision__: we have fixed it.

---

> ### Author Response · Authors · 2022-07-29
> **Response to the comments of Reviewer Q7gH Part II**
>
> ### Q1 When comparing the R-D curves of Code Editing Enhanced and Yang et al. [2020], what is the difference of the proposed method and is it reasonable?
> * In Appendix A.1 Fig A.1., we can see that our proposed approach achieve continuous rate control with little loss compared with [Yang et al. 2020] based on [Ballé et al. 2018] and [Minnen et al. 2018], and marginal R-D loss in very low/high bpp based on [Cheng et al. 2020]. This result is reasonable. We use a single decoder to achieve this while [Yang et al. 2020] require multiple decoder. It would be unreasonable if our method outperforms [Yang et al. 2020]. Again, we are neither superior nor inferior to [Yang et al. 2020], as the task is very different.
> * __For revision__: we have included the discussions.
> ### Q2 Why does combining semi-amortized inference and adaptive quantization step size improve when low-rate results?
> * As stated in Sec 2.3, the trainable $\Delta$ enables an extra parameter to the entropy model, which reduce the mismatched bitrate
> bitrate $E_{q(y|x)}[\log p_{θ_{λ_1}}(⌊y⌉)−\log p_{θ_{λ_0}}(⌊y⌉)]$. And the analysis in Sec 4.2 and Fig. 2 verifies this.
> ### Q3 Why are the low-rate results in Fig. 3 approaching baseline?
> * This is probably because the latent becomes too sparse in low bitrate region and the majority of them are very close to $0$ in the initial SGA. Despite the annealing temperature, this sparsity makes SGA difficult to generate samples other than $0$.
> ### Q4 what are the cases in which the authors' method does not work well
> * In general, our method does not work for the cases where encoding time matters, such as real-time communication. Our method is extremely useful for the cases where we encode just once but decode/view plural number of times, such as content delivery network.
> * __For revision__: we have included this in limitation.
> ### Reference
> * Y. Kim, S. Wiseman, A. Miller, D. Sontag, and A. Rush. Semi-amortized variational autoencoders. In International Conference on Machine Learning, pages 2678–2687. PMLR, 2018.
> *  Y. Yang, R. Bamler, and S. Mandt. Improving inference for neural image compression. Advances in Neural Information Processing Systems, 33:573–584, 2020.
> * L. Theis, W. Shi, A. Cunningham, and F. Huszár. Lossy image compression with compressive autoencoders. In 5th International Conference on Learning Representations, ICLR 2017, 2017.
> * M. Song, J. Choi, and B. Han. Variable-rate deep image compression through spatially-adaptive feature transform. In 2021 IEEE/CVF International Conference on Computer Vision, ICCV 2021, pages 2360–2369. IEEE, 2021.
> * J. Ballé, D. Minnen, S. Singh, S. J. Hwang, and N. Johnston. Variational image compression with a scale hyperprior. In International Conference on Learning Representations, 2018.
> * D. Minnen, J. Ballé, and G. D. Toderici. Joint autoregressive and hierarchical priors for learned image compression. Advances in neural information processing systems, 31, 2018
> * Z. Cheng, H. Sun, M. Takeuchi, and J. Katto. Learned image compression with discretized gaussian mixture likelihoods and attention modules. In Proceedings of the IEEE/CVF Conference on Computer Vision and Pattern Recognition, pages 7939–7948, 2020.

---

> > ### Comment · Reviewer_Q7gH · 2022-08-08
> > **Thank you for your detailed response**
> >
> > Thank you very much for your kind additional explanation. Your additional explanation and revision helped me understand your claim (Yang et al. 2020 requires multiple models, whereas the proposed method can achieve nearly the same RD-rate with a single model). I will raise my evaluation of the paper.

---

### Author Response · Authors · 2022-08-02
**Summary of Revision**

Thanks for your detailed review. We have uploaded the revised main text and supplementary material, with all the revisions marked in blue. Due to the space limitation, we could not include all the amendment in the main text. Below is a summary of revisions:

### Main Text
* Sec 2.1: We rewrite a more extensive formulation of the relationship between lossy NIC and VAE. (as suggested by kNRz)
* Sec 2.3: We modified the wrong expression of "differentiating cdf" into "taking the difference of cdf", and provide a stricter formulation on the discrete entropy model by formally distinguish pdf, cdf and pmf (as suggested by kNRz)
* Sec 2.5, 4.4: We rephrase "distortion-perception trade-off" into "multi-distortion trade-off". (as suggested by kNRz)
* Sec 2.6: We emphasis that the grid search is limited to quantization stepsize of $z$. (as suggested by JZkG)
* Sec 4.1: We emphasis that actual range encoding/decoding is used in results reported. (as suggested by kNRz)

### Appendix
* A.1: We add additional discussion on results of [Yang et al. 2020] and discuss the difference of our work compared with it. (as suggested by Q7gH)
* A.3: We add additional discussion on the difference between our quantization stepsize optimization and [Choi et al. 2019]. (as suggested by Q7gH)
* A.4. Go without Grid Search: We add a new section to discuss the impact of grid search and possibility of fixing the $\Delta_z=1$ & abandon grid search for speed. (as suggested by JZkG)
* A.5. Go beyond $bpp=1.0$: We add a new section to discuss the experimental results on very high bitrate ($bpp>1.0$). (as suggested by JZkG)
* B.2. We add additional results and discussion on segmentation based ROI results. (as suggested by JZkG)
* C.1. We add additional discussion on the scenarios where our method works/fails. (as suggested by Q7gH)

### Reference
* Y. Yang, R. Bamler, and S. Mandt. Improving inference for neural image compression. Advances in Neural Information Processing Systems, 33:573–584, 2020.
* M. Song, J. Choi, and B. Han. Variable-rate deep image compression through spatially-adaptive feature transform. In 2021 IEEE/CVF International Conference on Computer Vision, ICCV 2021, pages 2360–2369. IEEE, 2021.

---

### Meta-Review · Area_Chair_CBPw · 2022-08-23

**Recommendation:** Accept
**Confidence:** Certain

**Metareview:**

Thanks for your submission to NeurIPS.  The reviewers are all in agreement that the paper is ready for publication.  They in particular appreciated your rebuttals and changes to the paper, and increased their scores as a result.  The proposed method is novel, interesting, and performs well.

**Award:**

No

---

### Decision · Program_Chairs · 2022-09-14

Accept